# MoGA: Mixture-of-Groups Attention for End-to-End Long Video Generation

**Weinan Jia**[1][‡]**, Yuning Lu**[2][†][§]**, Mengqi Huang**[1][†]**, Hualiang Wang**[3][‡]**,**
**Binyuan Huang**[4][‡]**, Nan Chen**[1]**, Mu liu**[2]**, Jidong Jiang**[2]**, Zhendong Mao**[1]

[1] University of Science and Technology of China    [2] FanqieAI, ByteDance China
[3] Hong Kong University of Science and Technology    [4] Wuhan University
jiawn@mail.ustc.edu.cn, luyuning@bytedance.com, huangmq@ustc.edu.cn

## Abstract

Long video generation with Diffusion Transformers (DiTs) is bottlenecked by the quadratic scaling of full attention with sequence length. Since attention is highly redundant, outputs are dominated by a small subset of query–key pairs. Existing sparse methods rely on blockwise coarse estimation, whose accuracy–efficiency trade-offs are constrained by block size. This paper introduces Mixture-of-Groups Attention (**MoGA**), an efficient sparse attention that uses a lightweight, learnable token router to precisely match tokens without blockwise estimation. Through semantic-aware routing, MoGA enables effective long-range interactions. As a kernel-free method, MoGA integrates seamlessly with modern attention stacks, including FlashAttention and sequence parallelism. Building on MoGA, we develop an efficient long video generation model that end-to-end produces minute-level, multi-shot, 480p videos at 24 fps, with a context length of approximately 580k. Comprehensive experiments on various video generation tasks validate the effectiveness of our approach. Project website: `https://jiawn-creator.github.io/mixture-of-groups-attention/`.

## 1 Introduction

A growing body of research indicates that scaling laws are a primary driver of progress toward artificial general intelligence (Brown et al., 2020; Team et al., 2023; Kaplan et al., 2020). As model parameters and data scale to billions, transformer-based foundation models (Vaswani et al., 2017) often exhibit emergent capabilities (Wei et al., 2022; Kaplan et al., 2020; Radford et al., 2021). In video generation, given the inherently temporal nature, progress requires not only scaling parameters and data but, more critically, scaling the effective context length. This need is especially salient for long-form video generation (*e.g.*, movies), where persistent memory is essential for maintaining consistency of environments and characters (Yu et al., 2025).

The main challenge of vanilla attention (Vaswani et al., 2017) for long sequences is its computational cost, which grows quadratically with the context length. To mitigate the challenge, prior work (Zhuang et al., 2024; Tian et al., 2024; Huang et al., 2025b; Xiao et al., 2025; Wang et al., 2025a) adopts a multi-stage pipeline that first generates key frames and then synthesizes intermediate frames. However, this design yields disjoint objectives that are not directly optimized for the end task, leading to error accumulation across stages. It also introduces hand-crafted inductive biases, hindering scalability.

For *end-to-end* long video generation, one line of work compresses historical content to accommodate longer contexts (*e.g.*, via recurrent layers (Dalal et al., 2025) or FramePack (Zhang & Agrawala, 2025)), which inevitably results in information loss. A complementary direction exploits *sparse attention* (Zaheer et al., 2020) by restricting computation to a selected subset of salient query–key pairs. Existing selection strategies generally fall into two categories: (i) *static* selection, *i.e.*, prior-

---

[†]Corresponding authors.
[‡]Work done during internships at FanqieAI, ByteDance China.
[§]Project leader.

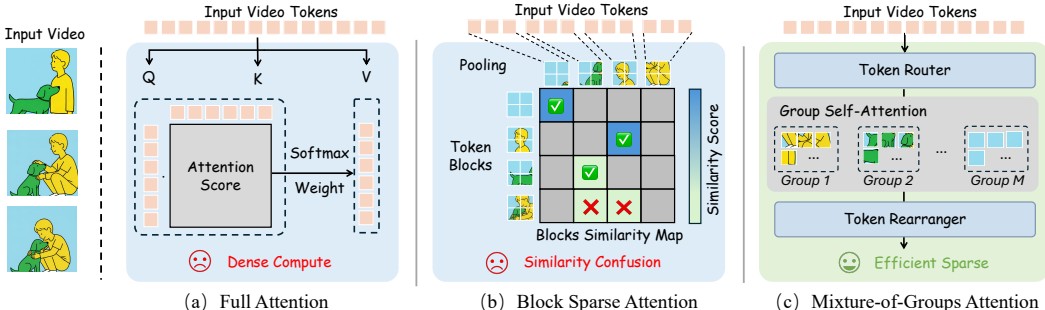

Figure 1: Illustration of our motivation. (a) Full attention suffers from dense computing when dealing with long sequences. (b) Block sparse attention may fail when block-level similarity is confused, resulting in unreliable attention. (c) Mixture-of-groups attention uses a lightweight token router (*i.e.*, a *single linear layer*) that assigns tokens to specialized groups, enabling groupwise attention and efficient long-context modeling.

driven heuristics that emphasize local spatiotemporal neighborhoods, which is efficient but limited in capturing dynamic long-range dependencies (Li et al., 2025; Xi et al., 2025; Gao et al., 2025; Seawead et al., 2025); and (ii) coarse-to-fine *dynamic* selection, which first estimates block-level important scores, routes query tokens to the top-k blocks, and then applies fine-grained attention within the selected blocks (Wu et al., 2025; Cai et al., 2025; Yang et al., 2025; Yuan et al., 2025; Lu et al., 2025). As shown in Fig.1 (b), the latter introduces an efficiency–performance trade-off: using larger blocks with a small top-k reduces the computational cost of the coarse stage but reduces selection performance.

In this work, we reveal that such coarse-grained estimation is unnecessary and each token should be precisely allocated. To achieve this, we propose Mixture-of-Groups Attention (**MoGA**), a simple and efficient dynamic token routing solution for end-to-end long video generation. A lightweight router (*i.e.*, a *single linear layer*) is employed to assign tokens to specific groups, as illustrated in Fig. 1(c), inspired by Mixture-of-Experts (MoE) (Jacobs et al., 1991). Full attention is then performed within each group, where the groupwise attention integrates seamlessly with modern attention kernels, *e.g.*, FlashAttention (Dao, 2023). Intuitively, the linear router's weight can be viewed as implicit cluster centers, enabling direct assignment of tokens to learnable anchors, without global similarity estimation. Furthermore, to balance long-range coherence and local fidelity, we couple MoGA with the spatiotemporal window attention (Gao et al., 2025), which can be considered as the groupwise attention with static and pre-defined groups. In addition, extended context alone is insufficient because a single global prompt cannot reliably control scene transitions or orchestrate events at precise time points in long videos. We therefore introduce shot-level textual conditioning via cross-modal attention, where each shot is guided by a concise description (Gu et al., 2025; Wang et al., 2025b). To support this, we build a data pipeline that produces minute-level video samples with dense, multi-shot captions and reliable shot segmentation.

Our contributions: We propose MoGA, an effective sparse attention mechanism that replaces block-level scoring with precise group assignment via a lightweight token router, enabling effective modeling of long contexts. Building on MoGA, we introduce a video generation model capable of producing minute-level, multi-shot, 480p videos at 24 fps with a context length of about 580k tokens. Extensive evaluations show consistent improvements over state-of-the-art (SoTA) sparse attention baselines and a multi-shot video generation model.

## 2 METHOD

In this section, we introduce MoGA for efficient long video generation. The overview of the whole architecture is shown in Fig. 2. We first present the preliminaries, then detail MoGA, and finally describe the pipeline for constructing multi-shot long-video training data.

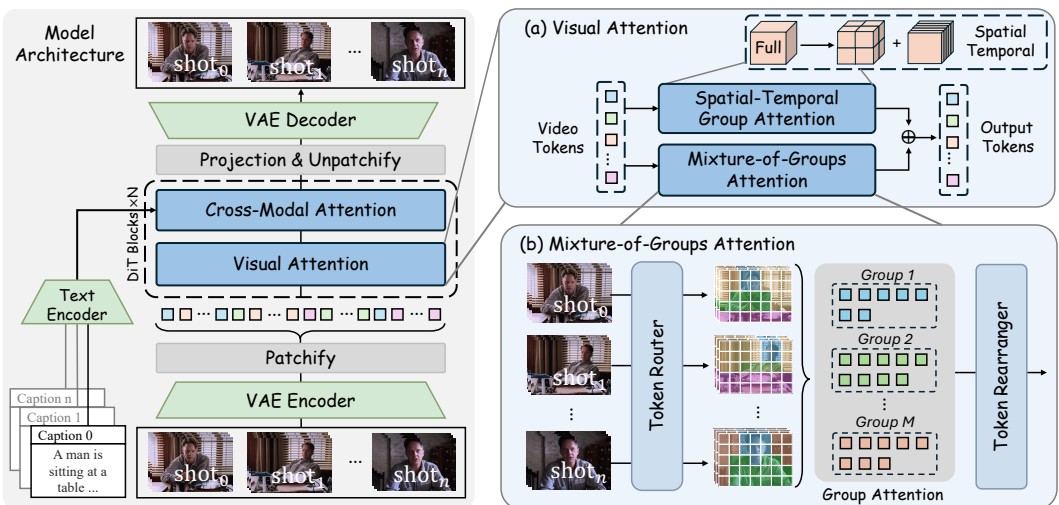

Figure 2: **Left**: Our model adopts a DiT architecture with interleaved Visual Attention and Cross-Modal Attention blocks. Visual Attention exclusively processes visual content, whereas Cross-Modal Attention enables shot-level text conditioning, instantiated via either cross-attention (Wan et al., 2025) or multi-modal attention (Kong et al., 2024; Esser et al., 2024). **Top-right** (a): Visual Attention combining MoGA with Spatial-Temporal Group Attention for global-local consistency. **Bottom-right** (b): MoGA, where a Router groups tokens and performs intra-group attention, enabling long-range global interactions.

## 2.1 PRELIMINARY

**Vanilla self-attention** Vaswani et al. (2017) plays a crucial role in video generation with Diffusion Transformer (DiT) Peebles & Xie (2023). Consider an input sequence $\boldsymbol{X} \in \mathbb{R}^{N \times d}$, where $N = h \times w \times t$ represents the total number of tokens across the latent spatial dimensions ($h \times w$) and the latent temporal dimension ($t$), with $d$ denoting the model's hidden dimension. For simplicity, we consider a single query case, where $\boldsymbol{x}$ is a token from the input sequence and $\boldsymbol{q}$ is its corresponding query. Vanilla self-attention (SA) is computed as:

$$\text{SA}(\boldsymbol{q}, \boldsymbol{K}, \boldsymbol{V}) = \text{softmax}(\frac{\boldsymbol{q}\boldsymbol{K}^\top}{\sqrt{d}}) \cdot \boldsymbol{V}, \quad (1)$$

where $\boldsymbol{K}$ and $\boldsymbol{V}$ denote the keys and values. While self-attention excels at capturing long-range dependencies via global information aggregation, it incurs a quadratic computational complexity of $\mathcal{O}(N^2)$. The computational burden becomes particularly prohibitive in long-video generation. For example, generating a 1-minute video at 480p with approximately 1,600 tokens per frame across 961 frames (16 fps) yields a total token count approaching 384k. Performing full-attention on such a long sequence is intractable.

Beyond computational cost, full attention is not ideally aligned with the structure of videos. In videos, softmax attention is inherently sparse Xi et al. (2025) because nearby tokens exhibit strong local spatiotemporal correlation, while only a few globally shared, dynamic semantics persist across frames. Most query–key pairs contribute little, whereas a small subset dominates Ge et al. (2023). For long videos, attention should leverage this sparsity by prioritizing important query–key interactions to reduce redundancy.

## 2.2 MIXTURE-OF-GROUPS ATTENTION (MOGA)

**MoGA** addresses the above challenge via efficient token routing, where a lightweight trainable router assigns correlated tokens to groups and performs self-attention within each group. Specifically, the router is a linear projection followed by softmax gating, similar to MoE Fedus et al. (2022). MoE scales model parameters by routing tokens to expert FFNs. In contrast, MoGA scales with respect to sequence length by modifying attention and routing tokens into different attention

---

Following Wan et al. (2025), VAE downsampling factors of (t/h/w) is (4,8,8) and patchify sizes is (1,2,2).

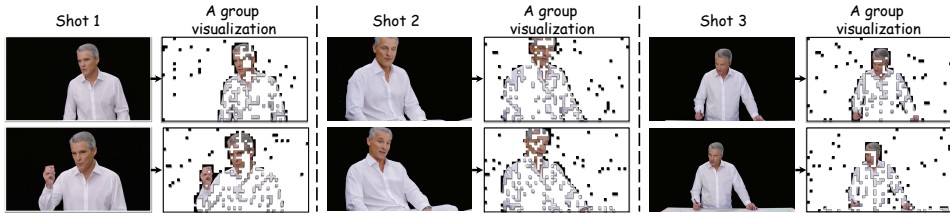

Figure 3: Visualization of dynamic router grouping. We visualize the visual patches of tokens from the same group assigned by an intermediate block's router. They focus on a specific visual concept across frames.

groups. Given a token $\boldsymbol{x} \in \mathbb{R}^d$ and a predetermined number of groups $M$, the router computes routing scores $\boldsymbol{r} \in \mathbb{R}^M$ as:

$$\boldsymbol{r} = \text{Router}(\boldsymbol{x}). \tag{2}$$

The group assignment probabilities are computed as:

$$p(i|\boldsymbol{x}) = \text{softmax}(\boldsymbol{r})_i, \tag{3}$$

and the token is assigned to the group with highest probability:

$$g(\boldsymbol{x}) = \arg \max_{i \in [M]} \ p(i|\boldsymbol{x}). \tag{4}$$

After grouping, we apply self-attention within each group independently. The output of MoGA is:

$$\text{MoGA}(\boldsymbol{x}) = p(g(\boldsymbol{x})|\boldsymbol{x}) \cdot \text{SA}(\boldsymbol{q}, \boldsymbol{K}_{g(\boldsymbol{x})}, \boldsymbol{V}_{g(\boldsymbol{x})}), \tag{5}$$

where $\boldsymbol{K}_{g(\boldsymbol{x})}$ and $\boldsymbol{V}_{g(\boldsymbol{x})}$ are the keys and values of the group $g(\boldsymbol{x})$, and $\boldsymbol{q}$ is the query feature of $\boldsymbol{x}$. This grouped attention mechanism reduces computational complexity from $\mathcal{O}(N^2)$ to a theoretical minimum of $\mathcal{O}(N^2/M)$ under uniform group assignment.

As illustrated in Fig. 3, we extract the grouping assignments from an intermediate-layer router during the video generation process and visualize one representative group. After end-to-end training, the router assigns the man's head, hands, and portions of his clothing to the same group, indicating its ability to capture semantically coherent structures that span shot boundaries.

MoGA builds on groupwise attention and remains compatible with high-performance kernels such as FlashAttention (Dao, 2023) (see Alg. 1). Beyond sparse attention, a second pillar of long-context modeling is sequence parallelism (Jacobs et al., 2023), with which MoGA is also compatible. Before the sequence gather and head scatter step in each attention layer, MoGA computes routing scores over tokens (with whole heads) and then aggregates the routing results across all tokens.

**Group Balancing Loss.** A potential issue with token assignment is that the router may collapse by routing most tokens to only a few groups, which would degrade MoGA into full attention. To encourage adaptive token allocation across groups, we introduce an auxiliary *group balancing loss*, inspired by the load balancing loss (Fedus et al., 2022) used in MoE. The loss is defined as:

$$\mathcal{L}_{\text{gb}} = \alpha \cdot M \cdot \sum_{i=1}^{M} F_i \cdot P_i, \tag{6}$$

where $\alpha$ is loss weight and $F_i$ is the fraction of tokens assigned to group $i$,

$$F_i = \frac{1}{N} \sum_{\boldsymbol{x}} \mathbf{1}(g(\boldsymbol{x}) = i), \tag{7}$$

where $\mathbf{1}$ is the indicator function, and $P_i$ is the mean routing probability allocated for group $i$,

$$P_i = \frac{1}{N} \sum_{g(\boldsymbol{x})=i} p(g(\boldsymbol{x})|\boldsymbol{x}). \tag{8}$$

Minimizing $\mathcal{L}_{\text{gb}}$ encourages uniform token assignment across groups, as this objective attains its minimum under a uniform distribution (Fedus et al., 2022).

**Spatial-Temporal Group Attention.** Although MoGA captures long-range coherence, it lacks local continuity. We complement it with a local spatiotemporal group attention (STGA) (Gao et al., 2025;

---

**Algorithm 1** MoGA Pseudocode with FlashAttention

---
1: $\boldsymbol{Q}, \boldsymbol{K}, \boldsymbol{V}$ are the query, key and value of tokens $\boldsymbol{X}$
2: $\boldsymbol{g} = \text{router}(\boldsymbol{X})$ ▷ MoGA routing results
3: $\hat{\boldsymbol{Q}}, \hat{\boldsymbol{K}}, \hat{\boldsymbol{V}}, \text{cu\_seq\_len}, \text{max\_seq\_len}, \text{permute\_index} = \text{permute}(\boldsymbol{Q}, \boldsymbol{K}, \boldsymbol{V}, \boldsymbol{g})$
4: $\hat{\boldsymbol{O}} = \text{flash\_attn}(\hat{\boldsymbol{Q}}, \hat{\boldsymbol{K}}, \hat{\boldsymbol{V}}, \text{cu\_seq\_len}, \text{max\_seq\_len})$
5: $\boldsymbol{O} = \text{repermute}(\hat{\boldsymbol{O}}, \text{permute\_index})$ ▷ MoGA recovers the original token positions

---

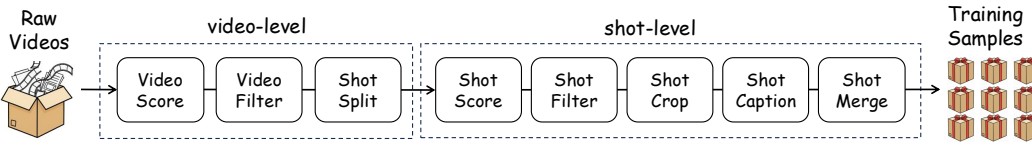

Figure 4: Multi-shot long video data pipeline.

Zhang et al., 2025c), which restricts self-attention to local windows in latent video space, as shown in Fig. 2(a). This captures short-range dependencies with bounded compute.

We first partition the latent video into fixed spatial windows and then group frames along the temporal axis. Frames from different shots are assigned to distinct temporal groups. We empirically find that completely removing inter-shot interactions causes flicker in the first frame after a shot cut. To mitigate this, when computing group attention, we augment the keys and values with two latent frames from adjacent shots (without augmenting the queries). This preserves continuity at shot boundaries with negligible additional compute. To enable intra-frame information exchange, we also perform per-frame attention by grouping tokens within each latent frame. Each token therefore receives outputs from multiple groups (one dynamic group and two static groups), and we take their mean as the final output.

### 2.3 DATA PIPELINE

We construct a pipeline that converts raw long videos into one-minute, multi-shot clips with dense annotations for long video generation. The pipeline has two stages: a video-level phase and a shot-level phase (Fig. 4).

**Video-level.** We first analyze raw videos using visual quality assessment (VQA) models (e.g., aesthetics (Schuhmann et al., 2022b), clarity, exposure) and simple operators (e.g., black-border detection) to obtain metadata and quality scores. We then filter raw videos with source-specific, calibrated thresholds to remove low-quality content. Because long-video samples requires temporally coherent, we relax clip-level filtering(Zheng et al., 2024; Kong et al., 2024) but apply stricter filtering at the source (raw-video) level. Next, we segment each video into single-shot clips, using AutoShot (Zhu et al., 2023) and PySceneDetect (Breakthrough & Contributors, 2014). AutoShot shows higher sensitivity to fades and gradual transitions. Combining both tools' predictions allow us to label whether a boundary is clean or affected by transition overlap. This stage yields a pool of single-shot clips.

**Shot-level.** We process single-shot clips using VQA and optical character recognition (OCR) models and discard low-quality clips. Based on OCR results, we compute a maximal-area crop that excludes watermarks and subtitles while preserving the original aspect ratio. Clips with insufficient retained area are discarded. Next, we generate captions of cropped clips by employing a multimodal large language model (Bai et al., 2025). Finally, we merge temporally adjacent single-shot clips into multi-shot training samples (up to 65 seconds) and trim a few frames from clips affected by transition overlap to ensure clean boundaries.

## 3 EXPERIMENTS

**Training settings.** We fine-tune MoGA on existing DiT-based short video generation models with the rectified flow objective Esser et al. (2024). For a fair comparison with baselines, we train MoGA on the open-source Wan2.1 models (1.3B and 14B) (Wan et al., 2025). The resulting model stably generates 477 frames at 16 fps (30-second) at 480p, with a context length of 187k. We use a constant learning rate of $1e$-5. The loss weight $\alpha$ is set to $0.1$. We set the number of groups to $M = 5$ and

| Method | Base Model | Subject Consistency ↑ | Background Consistency ↑ | Motion Smoothness ↑ | Aesthetic Quality ↑ | Image Quality ↑ | Text2Video CLIP ↑ | Overall Consistency ↑ | Temporal Flickering ↑ | Sparsity ↑ |
|---|---|---|---|---|---|---|---|---|---|---|
| Wan (Original) | Wan2.1-14B | 0.9611 | **0.9560** | **0.9936** | 0.5807 | 0.6680 | **0.2590** | **0.1855** | 0.9897 | 0% |
| DiTFastAttn (Training-based) | Wan2.1-14B | 0.9456 | 0.9394 | 0.9924 | 0.5269 | 0.6466 | 0.2461 | 0.1361 | **0.9899** | 50.00% |
| SVG (Training-free) | Wan2.1-14B | 0.9002 | 0.8926 | 0.9870 | 0.5370 | 0.6357 | 0.2516 | 0.1650 | 0.9714 | 50.00% |
| VMoBA (Training-free) | Wan2.1-14B | 0.8605 | 0.8876 | 0.9789 | 0.5369 | 0.6111 | 0.2492 | 0.1695 | 0.9523 | 31.00% |
| MoGA (Ours) | Wan2.1-1.3B | 0.9527 | 0.9462 | 0.9836 | 0.5519 | 0.6523 | 0.2502 | 0.1559 | 0.9721 | **71.25%** |
| MoGA (Ours) | Wan2.1-14B | **0.9699** | 0.9542 | 0.9927 | **0.5810** | **0.6994** | 0.2576 | 0.1743 | 0.9811 | **71.25%** |

Table 1: Quantitative comparison of 5-second single-shot video generation.

| Method | Base Model | Subject Consistency ↑ | Background Consistency ↑ | Motion Smoothness ↑ | Aesthetic Quality ↑ | Image Quality ↑ | Text2Video CLIP ↑ | Cross-Shots DINO ↑ | Cross-Shots CLIP ↑ |
|---|---|---|---|---|---|---|---|---|---|
| IC-Lora+Wan | Wan2.1-1.3B | 0.9476 | 0.9538 | 0.9901 | 0.5237 | 0.6684 | 0.2381 | 0.4669 | 0.7169 |
| Echoshot | Wan2.1-1.3B | 0.9544 | 0.9518 | **0.9939** | 0.5718 | 0.6534 | 0.2535 | 0.5961 | 0.8469 |
| MoGA (Ours) | Wan2.1-1.3B | **0.9549** | **0.9597** | 0.9919 | **0.5890** | **0.6729** | **0.2582** | **0.6623** | **0.8654** |
| MoGA (Ours) | Wan2.1-14B | 0.9651 | 0.9679 | 0.9954 | 0.5932 | 0.6867 | 0.2783 | 0.6703 | 0.8629 |

Table 2: Quantitative comparison of 10-second multi-shot video generation.

partition the spatial grid into $2 \times 2$ groups. We adopt a multistage training strategy: 3k steps on 10-second clips followed by 1k steps on 30-second clips.

Because MoGA is a general sparse attention, we also apply it to a video generation model built on MMDiT (Esser et al., 2024; Kong et al., 2024). Unlike Wan, this model replaces cross-attention with MMDiT to perform cross-modal attention. It partitions space into 4×4 groups and sets the router's group number to M = 20, enabling a much longer context length. This MMDiT-based model generates 1,441 frames at 24 fps (60-second) at 480p, with a context length of 578K.

**Baselines.** To evaluate our method, we compare with multiple baselines. For multi-shot long video generation, we include the keyframe-based pipeline IC-LoRA+Wan (Huang et al., 2024a; Wan et al., 2025) and EchoShot (Wang et al., 2025b), which natively supports multi-shot generation. For sparse video generation, we compared against sparse attention methods, including the training-based method DiTFastAttn (Yuan et al., 2024) and the training-free methods SVG (Xi et al., 2025) and VMOBA (Wu et al., 2025).

**Evaluation metrics.** Following prior work, we evaluate all methods using the metrics introduced by VBench (Huang et al., 2024b). Specifically, subject consistency and background consistency measure how well the main subjects and backgrounds of sampled frames are preserved throughout the video. Motion smoothness measures motion fluidity, penalizing jitter and abrupt transitions. We also report aesthetic quality and image quality to quantify the visual appeal and technical fidelity of each frame. Text2Video CLIP measures the semantic consistency between a generated video and its text prompt. Temporal Flickering reflects the temporal stability of videos. To compute cross-shot consistency, we first sample a fixed number of frames from different shots. We then compute feature similarities across shots using CLIP (Radford et al., 2021) and DINOv2 (Oquab et al., 2023) feature similarities across shots, referred to as Cross-Shot CLIP and Cross-Shot DINO. For single-shot 5-second video generation, we constructed a diverse test set comprising 300 prompts. For multi-shot 10-second video generation, we use the 100 multi-shot prompt sets, which introduced from (Wang et al., 2025b). We evaluate long video generation with a test set of 11 scripts comprising 105 prompts. Each script contains 8–10 shots to produce a 30-second video.

## 3.1 QUANTITATIVE RESULTS

First, we compare MoGA with prior sparse attention methods for single-shot, short video generation, following their evaluation settings to ensure fairness. As shown in Tab. 1, despite higher sparsity, MoGA achieves consistent improvements over existing sparse baselines across metrics. It is worth noting that although our method is highly sparse, it can still match or surpass original Wan (full attention) on multiple dimensions.

Next, we compare MoGA with other multi-shot video generation methods. Tab. 2 reports quantitative comparisons among MoGA, IC-LoRA+WAN, and EchoShot. Despite relying on sparse attention, our method outperforms the full attention baseline (EchoShot) on most metrics, indicating that preserving interactions among salient tokens not only reduces FLOPs but also suppresses noise from irrelevant content. This leads to stronger character identity consistency and improved temporal scene coherence.

| Method | Base model | Subject Consistency ↑ | Background Consistency ↑ | Motion Smoothness ↑ | Aesthetic Quality ↑ | Image Quality ↑ | Text2Video CLIP ↑ |
|---|---|---|---|---|---|---|---|
| IC-Lora+Wan | Wan2.1-14B | 0.8946 | 0.9169 | 0.9872 | 0.5759 | 0.6835 | 0.2547 |
| MoGA (Ours) | Wan2.1-1.3B | 0.9218 | 0.9204 | 0.9846 | 0.5731 | 0.6829 | 0.2579 |
| MoGA (Ours) | Wan2.1-14B | **0.9572** | **0.9475** | 0.9893 | 0.5789 | 0.6993 | **0.2634** |
| MoGA (Ours) | MMDiT | 0.9305 | 0.9301 | **0.9895** | **0.5881** | **0.6996** | 0.2614 |

Table 3: Quantitative comparison of 30-second multi-shot long video generation.

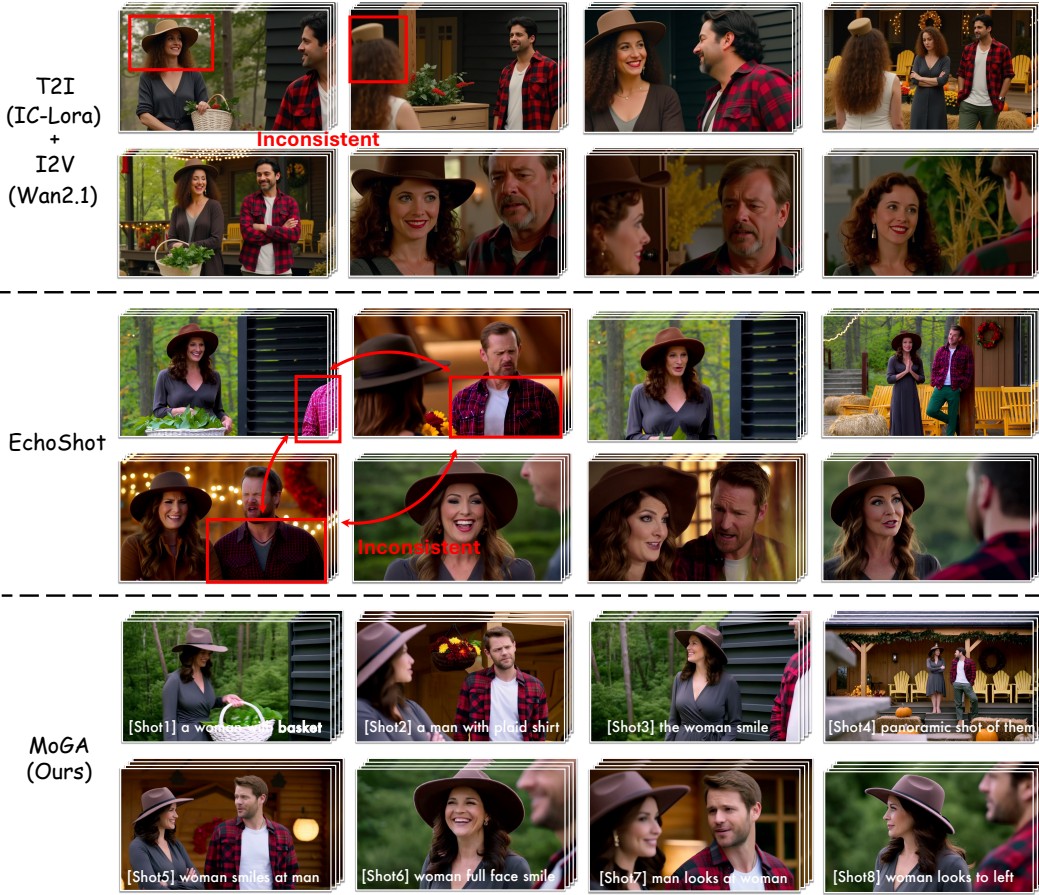

Figure 5: Qualitative of MoGA and other methods. We present eight representative shots, demonstrating long-range coherence, character consistency, and visual quality.

Finally, we benchmark long video generation against the baseline. Because few open-source methods can produce 30-second, multi-shot videos, we compare MoGA (with two backbones) to IC-LoRA+Wan. As shown in Tab. 3, MoGA substantially outperforms IC-LoRA+Wan under the same backbone, highlighting the benefits of end-to-end modeling over multistage pipelines. Notably, even under aggressive sparsity, MoGA with MMDiT maintains high visual fidelity, indicating a scalable path to longer context lengths.

## 3.2 QUALITATIVE RESULTS

In this subsection, we present qualitative results on 30-second videos across representative baselines. Because EchoShot cannot natively produce 30-second outputs, we concatenate video clips generated by EchoShot to form the full sequence. As shown in Fig. 5, the IC-LoRA+Wan pipeline is constrained by its per-iteration image cap (typically three frames), which limits its ability to cover a larger number of shots. Consequently, it often exhibits subject drift and background inconsistency as the sequence progresses. EchoShot scales to more shots but still shows notable cross-shot inconsistencies on long temporal distances. In contrast, MoGA maintains stable, coherent content

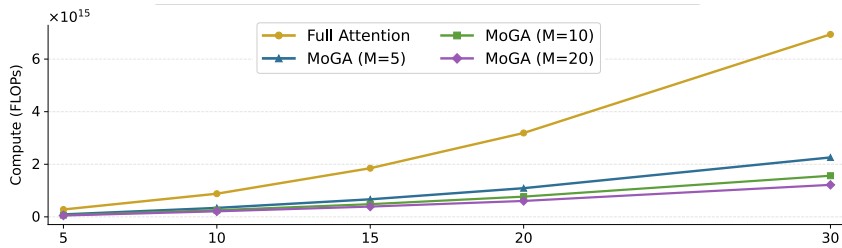

Figure 6: Computational efficiency. The x-axis denotes the generated video duration (s). As the number of groups increases, MoGA's Flops decrease substantially.

| Group num | Cross-shot CLIP ↑ | Cross-shot DINO ↑ | Sparsity | PFlops |
|---|---|---|---|---|
| 1 | 0.8206 | 0.5919 | 0% | 0.88 |
| 2 | 0.8589 | 0.6761 | 41.25% | 0.59 |
| 4 | **0.8672** | 0.6853 | 66.25% | 0.42 |
| 8 | 0.8606 | **0.6910** | 78.75% | 0.36 |
| 16 | 0.8569 | 0.6896 | 81.25% | 0.35 |

Table 4: Results of consistency for MoGA with Wan2.1-1.3B on 10s videos.

over extended durations. For example, even without repeated or explicit specification across shots, the woman's hat remains consistently preserved. Since STGA lacks explicit cross-shot information exchange, this consistency can be attributed to MoGA, which effectively selects and maintains shot-spanning identity and context. We provide more visualization in appendix A.1.

## 3.3 ABLATION STUDY

**Computational Efficiency.** Fig. 6 plots the relationship between the number of groups ($M$) and FLOPs for the Wan2.1-1.3B model. Our experiments show that even with a relatively small groups count $M = 5$ for 30-second videos, MoGA achieves substantial computational savings compared to full attention (2.26 PFlops vs. 6.94 PFlops). Meanwhile, it also delivers a $1.7\times$ speedup in both training and inference. Notably, unlike alternative sparse attention such as VMoBA, which incur additional memory overhead due to their block-based mechanisms, our approach maintains memory efficiency without additional memory consumption.

**Routing Group number M.** We conduct an ablation study on the number of groups under a fixed computational budget (in Tab. 4). Cross-shot DINO and CLIP scores exhibit a rise-then-fall trend as the number of groups increases. This suggests that a moderate level of grouped sparsity strikes a balance between global consistency and efficiency, yielding near-optimal consistency while maintaining computational efficiency.

**Effectiveness of MoGA and STGA.** As shown in Fig.7, MoGA and STGA play complementary roles in enabling context-consistent long video generation. Using MoGA alone lacks local information exchange and fails to produce meaningful visual content. Conversely, using only STGA limits long-range shot interactions, leading to poor cross-shot consistency and weakened narrative coherence. When combined, the model achieves strong cross-shot consistency. These results indicate that MoGA effectively routes and preserves shot-spanning identity and context at relatively low computational cost.

**Controllability of subject consistency.** Fig. 8 provides a comparison between MoGA and full attention. Both models are trained on 10-second data with Wan2.1-14B. The left panel illustrates MoGA's ability to maintain subject identity across multiple scenes, while the right panel demonstrates its robustness to appearance changes (e.g., clothing) when preserving identity consistency. Despite 71.25% sparsity, MoGA achieves narrative coherence and content editability on par with full attention, and in some cases delivers superior performance.

**Quantitative analysis of token routing.** We treat the router's token grouping as a form of unsupervised semantic segmentation and use SAM2 Ravi et al. (2024) to generate foreground masks for each video frame as ground truth (GT). For each GT mask, we match the group whose prediction yields the highest Intersection-over-Union (IoU) and take that IoU as the per-frame score. Final performance is reported as the average over different cases. Trained MoGA clearly surpasses random and untrained routers (28.6% vs. 15.6%/18.5%; Tab. 5). Over timesteps, IoU rises from 17.6% at

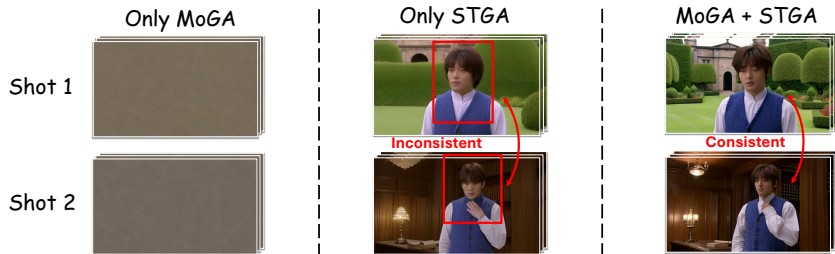

Figure 7: Visual ablation of MoGA and STGA.

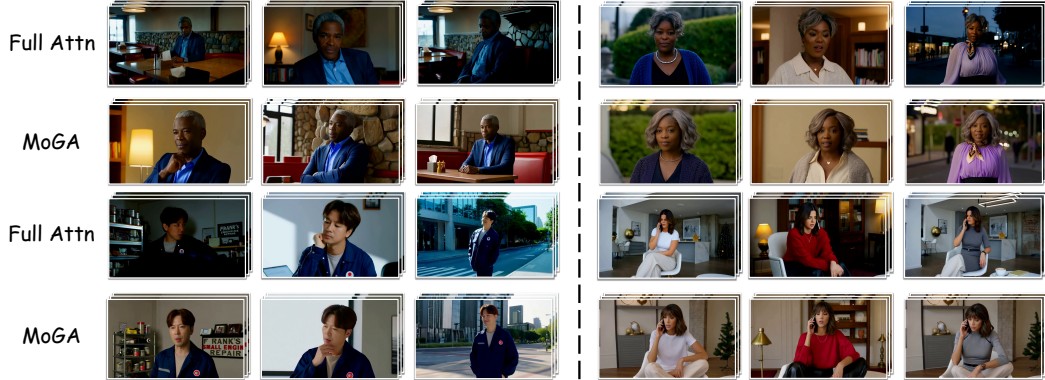

Figure 8: Visual comparison of MoGA vs. full attention on multi-shot generation with single subject. Left column shows the subject wearing the same outfit across different shots. Right column shows the subject changing outfits at shot transitions according to the text instructions.

t=999 to 28.6% and then stabilizes (Tab. 6). Across different DiT blocks, middle layers are strongest (up to 31.0%; Tab. 7). For more details, please refer to A.10.

## 4 RELATED WORK

### 4.1 LONG VIDEO GENERATION

Previous work on long video generation beyond typical duration limits has converged on three main paradigms. **Multistage** methods decompose long video generation into multiple steps (Yin et al., 2023; Zhuang et al., 2024; Tian et al., 2024; Huang et al., 2025b; Xiao et al., 2025; Wang et al., 2025a). For example, Captain Cinema (Xiao et al., 2025) adopts hierarchical planning with top-down keyframe generation and bottom-up synthesis for narrative coherence. Multistage approaches introduce hand-crafted inductive biases and pose challenges for end-to-end optimization. **Autoregressive** approaches generate videos through sequential segment synthesis (Chen et al., 2024; Huang et al., 2025c; Yin et al., 2025; Henschel et al., 2025; Gu et al., 2025; ai et al., 2025). Diffusion Forcing (Chen et al., 2024) adapts denoising schedules for variable sequence lengths. CasusVid (Yin et al., 2025) distills bidirectional models into an efficient autoregressive model. StreamingT2V (Henschel et al., 2025) combines short- and long-term memory for streaming video extension. FAR (Gu et al., 2025) introduces hierarchical causal representations for multiscale dependencies. MAGI-1 (ai et al., 2025) demonstrates the scaling capability of this paradigm. **Context compression** methods address computational constraints by compressing historical content (Dalal et al., 2025; Zhang & Agrawala, 2025; Jiang et al., 2025; Huang et al., 2025a). TTT (Dalal et al., 2025) compresses long context via a bidirectional recurrent layer. FramePack (Zhang & Agrawala, 2025) employs importance-based frame compression to maintain a fixed computational budget. Furthermore, M4V (Huang et al., 2025a) augments the original MMDiT blocks with a modified Mamba (Gu & Dao, 2024) tailored for multimodal and spatiotemporal modeling, achieving lower FLOPs and reducing training/inference latency while maintaining quality. However, these methods either produce videos of limited duration (Chen et al., 2024; Huang et al., 2025c; ai et al., 2025) or fail to generate multi-shot videos in real-world scenes (Yin et al., 2025; Henschel et al., 2025; Gu et al., 2025; Dalal et al., 2025; Zhang & Agrawala, 2025; Jiang et al., 2025). A closely related line of work

|  | Random | MoGA (Before Training) | MoGA (After Training) |
|---|---|---|---|
| IoU (%) | 15.6 | 18.5 | 28.6 |

Table 5: Comparison of IoU before and after training.

| Time Step | 999 | 956 | 931 | 884 | 853 | 499 | 438 | 364 | 272 | 155 |
|---|---|---|---|---|---|---|---|---|---|---|
| IoU (%) | 17.6 | 22.2 | 25.9 | 25.1 | 24.8 | 27.8 | 27.6 | 27.3 | 28.0 | 28.6 |

Table 6: Comparison of IoU across different time steps.

| Block Index | 1 | 5 | 10 | 15 | 20 | 25 | 30 | 35 | 38 |
|---|---|---|---|---|---|---|---|---|---|
| IoU (%) | 19.2 | 18.4 | 31.0 | 23.6 | 26.6 | 28.9 | 28.0 | 30.6 | 24.5 |

Table 7: Comparison of IoU across different blocks.

is LCT (Guo et al., 2025), which models interleaved multi-shot prompts and videos within a local context window using full attention. While pioneering end-to-end multi-shot long video generation, LCT remains constrained by the quadratic cost of full attention.

## 4.2 SPARSE ATTENTION FOR VIDEO GENERATION

Attention–based foundation models unify many domains and consistently exhibit a common sparsity structure (Lu et al., 2025; Yuan et al., 2025; DeepSeek-AI, 2025). In video generation, given the inherent sparsity, a natural approach to efficient generation is to select important query-key pairs. Prior work broadly falls into two categories: static priors (Zhang et al., 2025a; Xi et al., 2025; Li et al., 2025) and coarse-to-fine dynamic routing (Wu et al., 2025; Yang et al., 2025; Zhang et al., 2025b). Among static approaches, STA (Zhang et al., 2025a) employs 3D sliding windows with a hardware-aware implementation. SVG (Xi et al., 2025) uses online pattern selection to classify attention heads as spatial or temporal sparse attention. Radial-Attention (Li et al., 2025) uses a fixed attention mask whose sparsity grows with the query–key distance to perform spatiotemporal attention with $\mathcal{O}(n \log n)$ complexity. However, these methods struggle to modeling evolving long-range dependencies, which are crucial for maintaining cross-shot consistency. Moreover, the implementation of fixed mask (*e.g.*, Radial-Attention) requires $\mathcal{O}(n^2)$ memory, which is prohibitive for long videos (300 GB memory for a one-minute video). MoGA avoids the mask-based design, making it practical at long context. Another line of work adopts dynamic token routing for sparse attention. VSA (Zhang et al., 2025b) first obtains compressed representations of contiguous spatiotemporal blocks, and then selects the top-k blocks for fine-grained attention. Similarly, VMoBA (Wu et al., 2025) extends the idea of MoBA (Lu et al., 2025) to video with tailored block structures and threshold-based selection. In such methods, the block size presents a trade-off between expressiveness and efficiency. Smaller blocks yield more accurate coarse-grained attention estimates but reduce efficiency. In addition, SVG2 (Yang et al., 2025) is a training-free dynamic sparse attention method that performs online k-means clustering over tokens during inference and selects the top-k clusters based on their centroids. It shares a similar motivation with MoGA, i.e., tokens can be grouped into semantically coherent clusters. However, online clustering in SVG2 introduces additional k-means computations during the forward pass and is not straightforward to differentiate through. In contrast, MoGA employs trainable cluster centroids to enable simple and efficient routing with minimal computational overhead, making it suitable for end-to-end training.

## 5 CONCLUSION

This paper introduces MoGA, a sparse attention that replaces coarse block-level scoring with precise, learned group assignments via a lightweight token router. By routing tokens into coherent groups, MoGA improves attention efficiency and fidelity for very long context. Building on MoGA, we propose the video generation model that produces minute-level, multi-shot videos at 480p and 24 fps. Diversity experiments on video generation demonstrate the effectiveness of our approach.

## 6 ACKNOWLEDGMENT

This research is supported by National Natural Science Foundation of China under Grant 623B2094. We thank the ByteDance Seedance team and Wenfeng Lin for their support.

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

## A APPENDIX

### A.1 ONE-MINUTE VIDEO OF 1,441 FRAMES

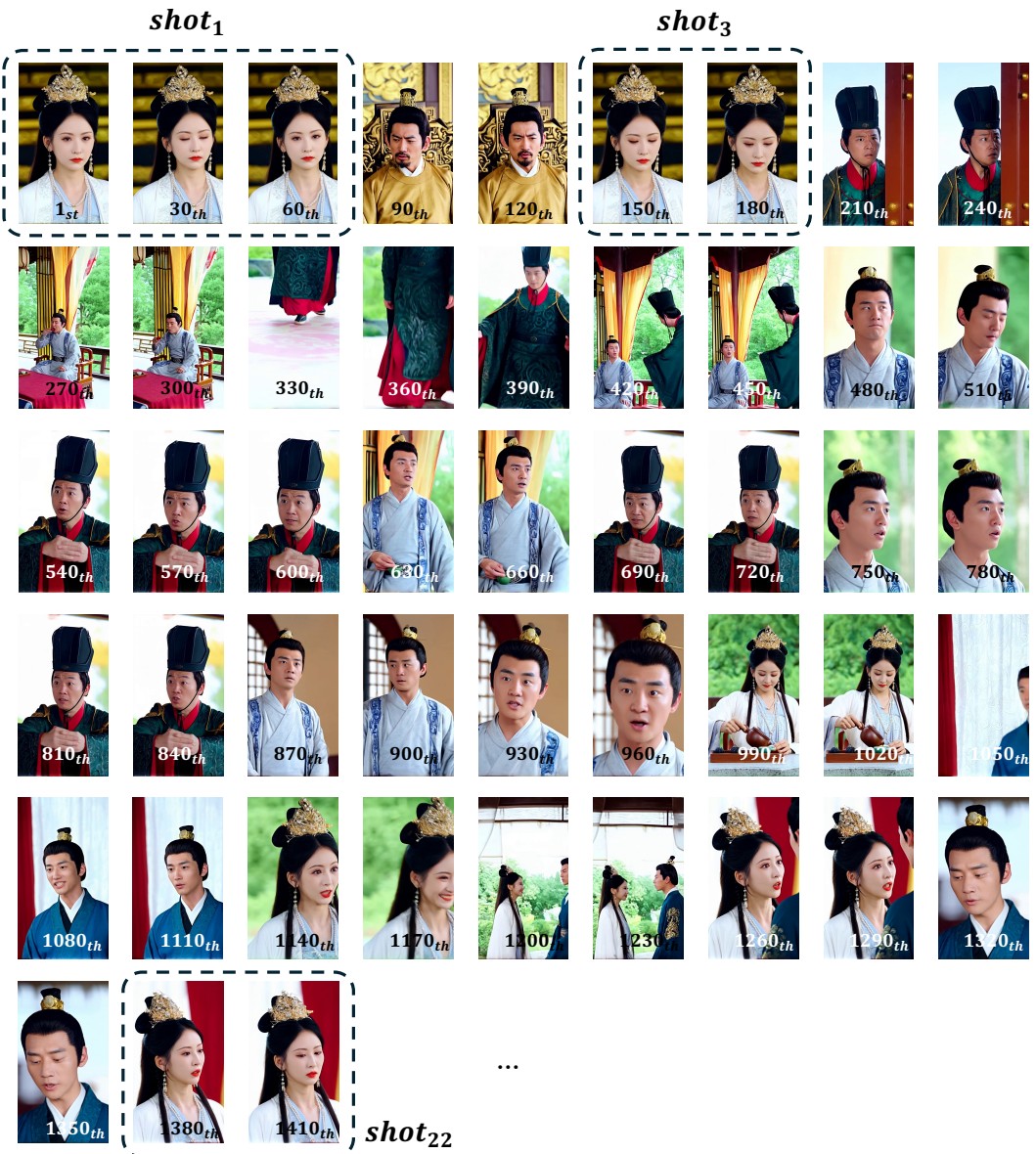

Figure 9: One-minute video generated by MoGA.

As shown in Fig. 9, we present the generated results of MoGA on an ultra-long video exceeding one minute, , using the MMDiT-based MoGA model ($M = 20$). MoGA maintains strong long-range contextual consistency. The 1st and 22nd shots remain highly coherent, and fine details such as the woman's hairpin and earrings are preserved across shots. Moreover, even with multiple faces appearing across different shots, the model avoids identity confusion.

### A.2 EMERGENCE OF BACKGROUND CONSISTENCY

As shown in Fig. 10, we demonstrate MoGA's ability to maintain background consistency. After training on long, multi-shot videos, MoGA exhibits emergent, implicit control over consistency in both the environment and the characters. Even without explicit specification of details (e.g., the

cabinet shape and the position of intravenous drip bottle), different shots automatically maintain coherent, temporally consistent depictions.

Figure 10: Emergence of background consistency.

## A.3 MULTI-STYLE VIDEO GENERATION

Fig. 11 illustrates MoGA's multi-style generation capability. MoGA not only performs strongly in realistic spaces, but also excels in stylized domains such as animation, illustration, and cinematic aesthetics. It can produce high-quality long 2D videos and long 3D videos while maintaining temporal coherence, identity consistency, and scene continuity across diverse styles and camera motions.

## A.4 DETAILS OF THE COMPUTATIONAL COMPLEXITY

As shown in Tab. 8, it reports the computational cost under varying numbers of groups ($M$) and video duration. As the generation video duration increases, the computational complexity of STGA exhibits approximately linear growth and the computational complexity of MoGA is approximately $1/M$ of that of Full Attention.

## A.5 ANALYSIS OF GROUP BALANCING LOSS

As shown in Fig. 12, we validate the effectiveness of the group balancing loss, which measures the balance of the router's token-to-group assignments. A higher value indicates that tokens concentrate

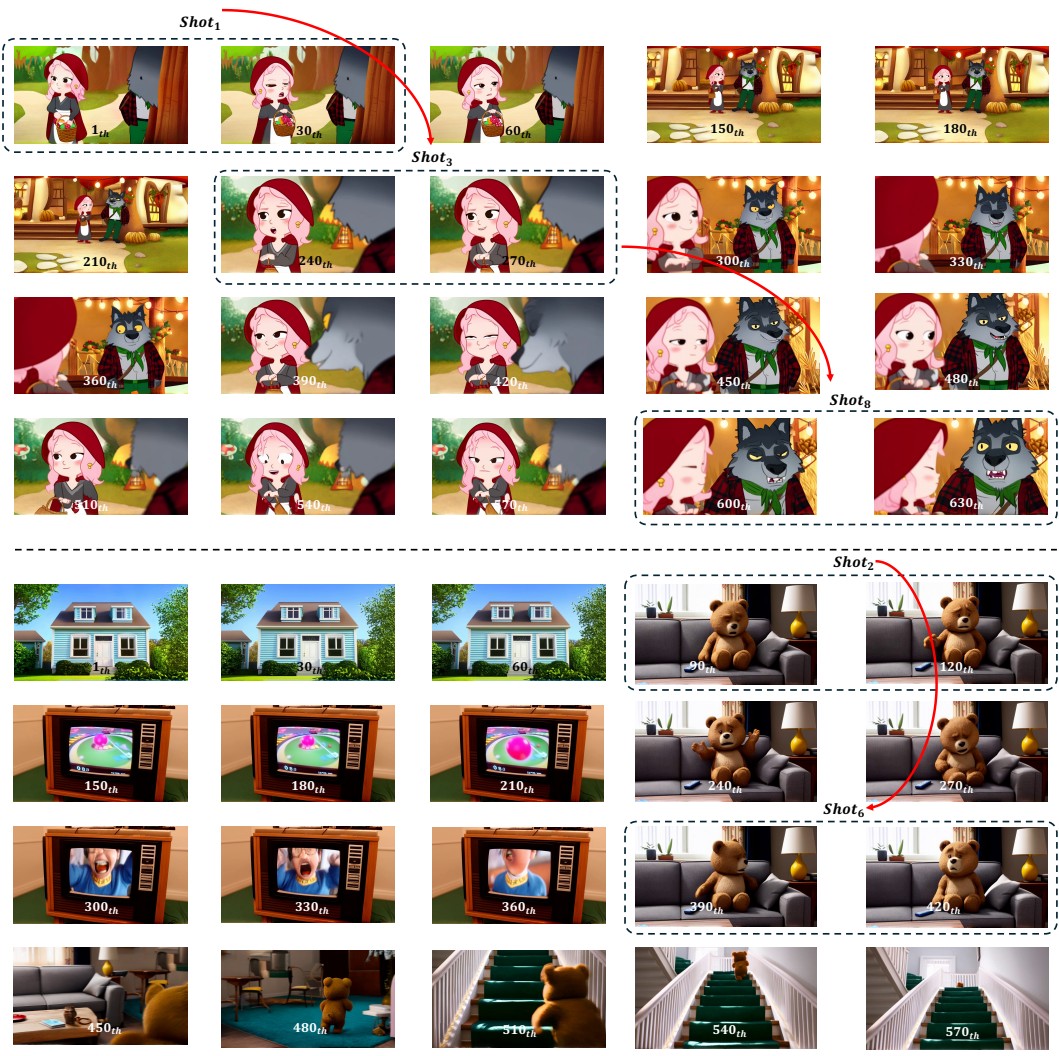

Figure 11: Multi-style video generation of MoGA.

| | Video Duration | 5s | 10s | 15s | 20s | 30s |
|---|---|---|---|---|---|---|
| | Frames | 77 | 157 | 237 | 317 | 477 |
| | Sequence length | 31200 | 62400 | 93600 | 124800 | 187200 |
| PFLOPs | Full Attention | 0.28 | 0.88 | 1.85 | 3.19 | 6.94 |
| | MoGA (M=5) | 0.093 | 0.34 | 0.67 | 1.09 | 2.26 |
| | MoGA (M=10) | 0.065 | 0.25 | 0.48 | 0.78 | 1.56 |
| | MoGA (M=20) | 0.051 | 0.21 | 0.39 | 0.61 | 1.22 |

Table 8: Compute (PFLOPs) versus group number $M$ and video duration on Wan2.1-1.3B.

in a few groups, whereas a lower value indicates more balanced grouping. When we include this loss during training, the metric rapidly converges to around 1, reflecting globally balanced assignments. In contrast, without it, the metric increases as the router funnels tokens into a few groups to gain short-term advantages in the diffusion MSE loss. Because our goal is to separate weakly related tokens and maintain balanced grouping, the additional group balance loss is necessary to enforce the desired assignments.

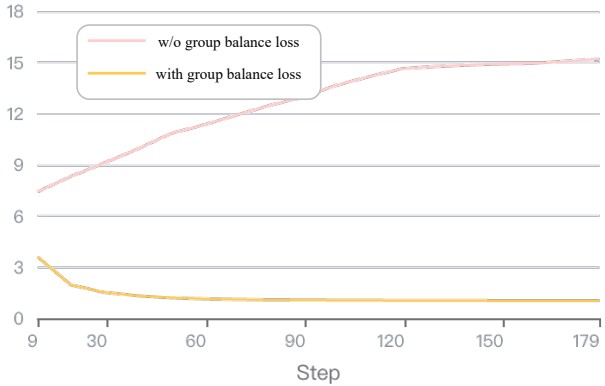

Figure 12: Group balancing loss curves of MoGA.

### A.6 DETAILED PYTORCH PSEUDO-CODE

```python
def moga_permute(q, k, v, router_logits, T=1):
    weights = (router_logits/T).softmax(dim=-1)
    weights, groups = torch.topk(weights, k=1, dim=-1)
    weights, groups = weights.reshape(-1), groups.reshape(-1)
    sort_idx = torch.argsort(groups, stable=True)
    groups_sorted, weights_sorted = groups[sort_idx], weights[sort_idx]
    _, counts = torch.unique_consecutive(
        groups_sorted, return_counts=True)
    cu_seqlens = torch.cat([counts.new_zeros(1), counts.cumsum(0)])
    q, k, v = q[sort_idx], k[sort_idx], v[sort_idx]
    return q, k, v, sort_idx, weights_sorted, cu_seqlens

def moga_repermute(out, sort_idx, weights_sorted):
    out = out * weights_sorted[:, None, None]
    out_ = torch.zeros_like(out)
    out_.index_add_(0, sort_idx, out)
    return out_

def mixture_of_groups_attention(q, k, v, router_logits):
    '''
    q, k, v: (L, H, D)
    router_logits: (L, M)
    '''
    q, k, v, sort_idx, weights_sorted, cu_seqlens = moga_permute(
        q, k, v, router_logits)
    max_seqlen = cu_seqlens.max().item()
    out = flash_attn_varlen_func(q, k, v, cu_seqlens, max_seqlen)
    out = moga_repermute(out, sort_idx, weights_sorted)
    return out
```

We provide a minimal implementation of MoGA based on PyTorch.

### A.7 EFFECT OF THE ROUTER TEMPERATURE

As shown in Fig. 13 and the quantitative analysis in Tab. 9, we adjust the router temperature hyperparameter and find that cross-shot similarity does not exhibit significant change, and there is no obvious impact on the router's stability.

### A.8 EFFECT OF THE GROUP BALANCING LOSS WEIGHT

As shown in Fig. 14 and Tab. 10, when we increase the balancing loss weight $\alpha$ to 1.0, the loss does not exhibit significant fluctuations, but the cross-shot consistency slightly decreases.

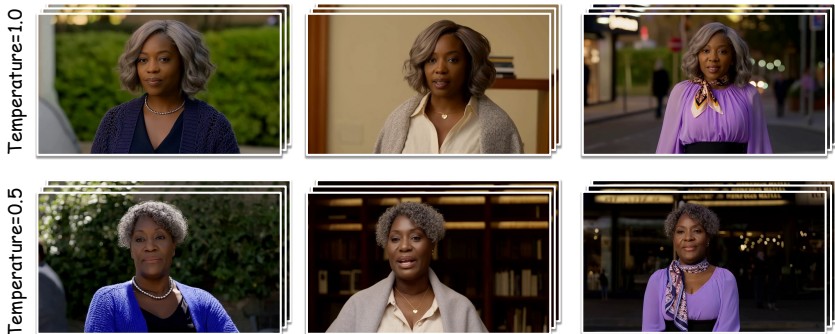

Figure 13: Visualization results under different temperatures.

| Temperature | Cross-shot DINO | Cross-shot CLIP |
| --- | --- | --- |
| 1 | 0.7284 | 0.8970 |
| 0.5 | 0.7286 | 0.8946 |

Table 9: Results under different temperatures.

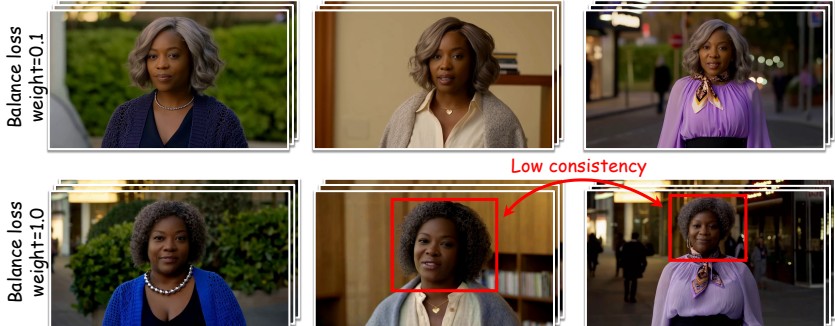

Figure 14: Visualization results under different balance loss weights.

| $\alpha$ | Cross-Shot DINO | Cross-Shot CLIP |
| --- | --- | --- |
| 0.1 | 0.7284 | 0.8970 |
| 1.0 | 0.7157 | 0.8851 |

Table 10: Results under different balance loss weights.

## A.9    VISUALIZATIONS OF TOKEN ROUTING

We select a specific group from a router (the first group in the 14th DiT block) to examine the relationship between token assignment and the corresponding visual patches. We visualize the visual patches corresponding to this group of tokens.

As shown in Fig. 15, we demonstrate that the router develop consistent semantic specializations for the visual concept without explicit supervision. For instance, this selection group consistently capture face-related tokens across diverse video samples.

## A.10    QUANTITATIVE ANALYSIS OF TOKEN ROUTING

We develop a quantitative analysis tool to measure whether the router assigns semantically related tokens to the same group. We treat the router's grouping of tokens a form of unsupervised segmentation. We then use SAM2 (Ravi et al., 2024) to obtain foreground masks for each frame as ground truth (GT), and treat the tokens aggregated by each group as predictions. For each GT mask, we match the group whose prediction mask achieves the highest IoU, and use the IoU as the metric.

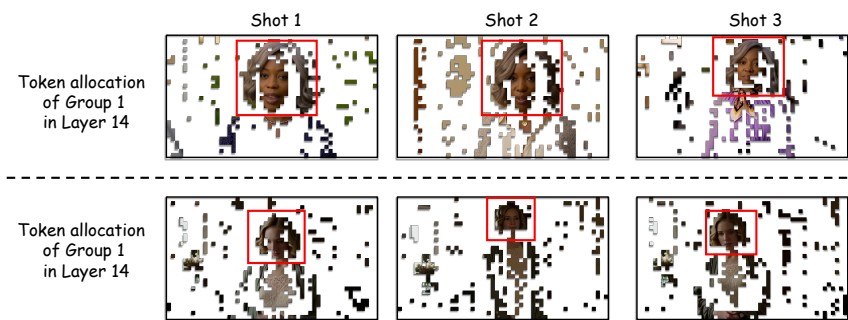

Figure 15: More visualization results of token routing.

| Duration | Seq Len | Full Attention | MoGA (M=5) | Speedup (M=5) | MoGA (M=20) | Speedup (M=20) |
|---|---|---|---|---|---|---|
| 5 s | 31,200 | 40.06 ms | 27.43 ms | 1.46× | 21.58 ms | 1.86× |
| 10 s | 62,400 | 158.83 ms | 81.15 ms | 1.96× | 62.24 ms | 2.55× |
| 20 s | 124,800 | 650.08 ms | 231.81 ms | 2.80× | 151.16 ms | 4.30× |
| 30 s | 187,200 | 1,423.52 ms | 455.57 ms | 3.12× | 267.24 ms | 5.33× |

Table 11: Latency comparison between Full Attention and MoGA.

We use the average IoU of different prompts as the final result. For evaluation, we use 9 scripts containing 27 prompts to generate 10-second videos. The IoU reflects the router's ability to assign tokens of the same category to a single group.

**Train vs. Random**. As shown in Tab. 5, we compare three routing methods: random assignment, a randomly initialized router, and the trained router. After training, specific groups achieve substantially higher IoU (28.6%) than both the random baseline (15.6%) and the randomly initialized router (18.5%).

**Different Time Step**. As shown in Tab. 6, we evaluate the IoU of the router at different sampling steps during inference. The router assigns tokens suboptimally (low IoU, 17%) at the initial denoising step (t=999). As denoising progresses, the IoU quickly rises above 25% (t=931) and remains stable, gradually increasing to 28.6%, which indicates that MoGA is robust in the semantics of token assignments throughout the denoising process.

**Different Blocks**. As shown in Tab. 7, we evaluate the IoU of the router at different DiT blocks during inference. The middle DiT blocks exhibit relatively stronger semantic grouping ability (IoU up to 31.0%), which is consistent with prior work indicating that intermediate layers of DiT can capture high-level semantic features (Yu et al., 2024).

## A.11 RUNTIME AND MEMORY USAGE

We compare MoGA (with STGA) and full attention (FlashAttention) in operator runtime, end-to-end training/inference time (sequence parallel=8; duration=30 s), and memory usage. All tests base on the same hardware, Ascend 910B NPU (TFLOPs roughly comparable to NVIDIA A100; 64 GB memory).

**Operator Comparisons (Latency)**. As shown in Tab. 11, MoGA consistently outperforms Full Attention across different sequence lengths, with larger gains at longer durations. Speedup grows with sequence length: from 1.46×–1.86× at 5 s to 3.12×–5.33× at 30 s.

**End-to-End Comparisons**. For 30s training dataset and Wan2.1-14B, the wall-clock times are measured in Tab. 12. MoGA significantly reduces both training and inference time versus full attention, with gains at M=5 (1.72× train, 2.10× inference) and M=20 (2.24× train, 2.43× inference).

## A.12 USER STUDY

We conduct a user study. Each user rates the generated videos (1-10), with rating dimensions including three aspects: video quality, consistency, and prompt following. The study covers 10s and 30s

|  | Full Attention | MoGA (M=5) | MoGA (M=20) |
|---|---|---|---|
| Training time (per iter) | 66.87 s | 38.76 s | 29.84 s |
| → Speedup vs. Full | — | 1.72× | 2.24× |
| Training RAM | 48.1 GB | 49.8 GB | 49.6 GB |
| Inference time (per step) | 40.21 s | 19.06 s | 16.55 s |
| → Speedup vs. Full | — | 2.10× | 2.43× |
| Inference RAM | 33.4 GB | 38.6 GB | 38.6 GB |

Table 12: Training and inference efficiency comparison.

| Metric (↑ better) | EchoShot | IC-LoRA+Wan | MoGA |
|---|---|---|---|
| Prompt Following | 6.97 | 5.58 | 8.47 |
| Video Quality | 7.11 | 4.53 | 8.05 |
| Consistency | 6.37 | 5.03 | 8.26 |

Table 13: Comparison across metrics.

multi-shot scripts and conducts a blind comparison among MoGA, EchoShot, and IC-LoRA+Wan. As shown in Tab. 13, MoGA clearly outperforms the baselines across all metrics, achieving the highest scores in Prompt Following (8.47), Video Quality (8.05), and Consistency (8.26). Compared with EchoShot, MoGA shows notable gains (+1.50, +0.94, +1.89 respectively), and it decisively surpasses IC-LoRA+Wan with even larger margins (+2.89 to +3.73).

## A.13 THE DETAILS OF DATA PIPELINE

We first collect publicly available long-form videos, including movies, TV series, animations, and short dramas. We then filter out videos with resolution below 720p, duration shorter than 1 minute, and videos whose VQA scores (sparse sampling) falls below the thresholds (e.g., the aesthetic threshold (Schuhmann et al., 2022a) ranges from 4 to 4.5 depending on the source). Next, we employ AutoShot and PySceneDetect jointly to segment the raw videos, obtaining single-shot clips with clean boundary transitions. After that, we perform denser frame sampling for each clip and compute per-clip VQA scores as well as OCR results. Based on OCR, we crop each clip while preserving the original aspect ratio to remove watermarks and subtitles. Clips whose cropped area is less than 40% of the original or whose VQA scores are below the thresholds are discarded. Finally, we concatenate clips from the same source video in chronological order and use an MLLM to caption each clip, assembling multi-shot long video training samples. We thus obtain approximately 200k training samples.

## A.14 LLM USAGE

Large Language Models (LLMs) were employed exclusively to assist with manuscript preparation. Their role was limited to refining language, improving readability, and enhancing the clarity of exposition. Specifically, the LLM contributed to tasks such as rephrasing sentences, correcting grammar, and smoothing the overall flow of the text.

The LLM played no part in formulating research questions, designing methodologies, conducting experiments, or analyzing results. All scientific ideas, experimental designs, and analyses were conceived and executed entirely by the authors. The authors accept full responsibility for the manuscript's content, including any text edited with LLM assistance. Care was taken to ensure that LLM-derived text complies with ethical standards and does not introduce plagiarism or scientific misconduct.

