# OpenReview forum: "MoGA: Mixture-of-Groups Attention for End-to-End Long Video Generation"
_ICLR.cc/2026/Conference — ICLR 2026 Poster_

### Official Review · Reviewer_ANEs · 2025-10-29

**Soundness:** 3
**Presentation:** 3
**Contribution:** 2
**Rating:** 6
**Confidence:** 5

**Summary:**

This paper introduces Mixture-of-Groups Attention (MoGA), a sparse attention mechanism designed to alleviate the quadratic complexity of Diffusion Transformers in long video generation. MoGA employs a lightweight, learnable token router to assign tokens into semantically coherent groups and performs full attention within each group, achieving fine-grained dynamic sparsification. Coupled with local spatiotemporal window attention (STGA) and shot-level textual conditioning, the proposed framework can end-to-end generate 480p, 24 fps, minute-long multi-shot videos. The authors also describe a multi-stage data curation pipeline and evaluate MoGA across single-shot, multi-shot, and ultra-long video scenarios.

**Strengths:**

* Originality: The MoGA design brings a Mixture-of-Experts–style router into sparse attention for video generation, avoiding coarse block-level heuristics that impede prior methods. Its compatibility with FlashAttention and sequence parallelism underscores solid systems innovation.
* Quality: Experiments cover single-shot short videos, multi-shot videos, and long-form generation, measured by VBench metrics (subject/background consistency, motion smoothness, aesthetics, etc.) plus Cross-Shot CLIP/DINO. MoGA outperforms both sparse and full-attention baselines, and the ablations (group count, STGA interplay, balancing loss) lend credibility.
* Clarity: The manuscript is well-structured, with clear figures contrasting full, block-sparse, and MoGA attention. Pseudocode and complexity analysis clarify implementation.
* Significance: Delivering stable 580k-token, minute-long, multi-shot videos is a notable leap for open-source systems, and MoGA’s design could generalize to other long-sequence tasks.

**Weaknesses:**

* Router design details: While a group balancing loss is introduced, the paper does not quantify how router collapse affects performance or explore router hyperparameters (dimension, temperature, regularization) beyond group count.
* Compute/resource reporting: PFLOP estimates and speedup claims are provided, yet concrete wall-clock measurements (training/inference time, memory usage) on specific hardware are missing, hindering reproducibility assessments.
* Data pipeline reproducibility: The multi-stage filtering/captioning pipeline depends on various proprietary tools (VQA/OCR/AutoShot), but code or parameter details are sparse. More statistics on filtered dataset size/quality would help.
* Limited multimodal control evaluation: Although shot-level textual conditioning is highlighted, there is little quantitative evidence (e.g., prompt adherence metrics or user studies) demonstrating textual control effectiveness.

**Questions:**

* Effectiveness of router: If the router only classify the tokens based on itself, would it be less sensitvie to global related information? Is there any detailed visusalization about router?
* Router stability: Did you observe router collapse or gradient instability early in training? Is the group balancing coefficient α = 0.1 robust across model sizes? How do performance and sparsity change if α varies?
* Choosing the number of groups: Since too few or too many groups hurt consistency, have you explored dynamically adjusting group counts (e.g., by sequence length or diffusion timestep)? Could this enhance performance or stability?
* Multimodal control evaluation: Can you provide quantitative metrics (e.g., text-to-video CLIP scores, prompt coverage, human preference studies) to support the claimed shot-level textual orchestration?
* Resource requirements: Please detail the hardware setup (GPU type/count), training time, and memory footprint for MoGA at M = 20 on minute-long videos so the community can gauge reproducibility.
* Broader applicability: Have you experimented with MoGA on non-video long-sequence tasks (e.g., text generation, 3D scene modeling)? If not, what considerations would be necessary for such extensions?

---

> ### Author Response · Authors · 2025-11-26
> **Author Response to Reviewer ANEs (Part 1/2)**
>
> Thank you for the thoughtful feedback and valuable suggestions. Below we respond point-by-point.
>
> > **Q1: Effectiveness of router**
>
> MoGA can be regarded as a method that directly aggregates semantically related tokens globally via a learnable router. Tab. 3 visualizes tokens of specific groups, which focus on the particular visual concept. We *add* more visualizations in Appendix A.9 of the revised version.
>
> Furthermore, we develop a *quantitative analysis tool* to assess whether the router assigns semantically related tokens to the same group. We discuss this below and integrate the methodology and results in Appendix A.10.
>
> We regard the router’s grouping of tokens as a form of unsupervised segmentation. We then use SAM2 to obtain foreground masks for each frame as ground truth (GT) and treat the tokens aggregated by each group as predictions. For each GT mask, we match the group whose predicted mask achieves the highest Intersection over Union (IoU), and use the IoU as the metric.
> We report the average IoU across different prompts as the final result. The IoU measures the router’s ability to assign tokens of the same category to a single group.
>
> * **Train vs. Random**: We compare three routing methods: random assignment,a randomly initialized router, and the trained router. After training, specific groups achieve substantially higher IoU (28.6%) than both the random baseline (15.6%) and the randomly initialized router (18.5%).
>
>     | Methods |IoU (%)|
>     |:---|:---:|
>     | Random | 15.6|
>     |MoGA (Before Training)|18.5|
>     |MoGA (After Training)|28.6 |
>
> * **Different Timesteps**: We evaluate the IoU of the router at different sampling steps during inference. The router assigns tokens suboptimally (low IoU, ~17%) at the initial denoising step (t=999). As denoising progresses, the IoU quickly rises above 25% (t=931) and remains stable, gradually increasing to 28.6%, which indicates that MoGA is robust in maintaining the semantic consistency of token assignments throughout the denoising process.
>
>     | Time Step | 999 | 956 | 931 | 884 | 853 | 499 | 438 | 364 | 272 | 155 |
>     |:---|:---:|:---:|:---:|:---:|:---:|:---:|:---:|:---:|:---:|:---:|
>     | IoU (%) | 17.6 | 22.2 | 25.9 | 25.1 | 24.8 | 27.8 | 27.6 | 27.3 | 28.0 | **28.6** |
>
> * **Different Blocks**: We evaluate the IoU of the router at different DiT blocks during inference. The middle DiT blocks exhibit relatively stronger semantic grouping ability (IoU up to 31.0%), which is consistent with prior work indicating that intermediate layers of DiT can capture high-level semantic features.
>
>     | Block Index | 1 | 5 | 10 | 15 | 20 | 25 | 30 | 35 | 38 |
>     |:---|:---:|:---:|:---:|:---:|:---:|:---:|:---:|:---:|:---:|
>     | IoU (%)| 19.2 | 18.4 | 31.0 | 23.6 | 26.6 | 28.9 | 28.0 | 30.6 | 24.5 |
>
>
> > **Q2: Router stability**
>
> We do *not* observe router collapse or gradient instability early in training. The group balancing loss converges rapidly after tens of steps, as shown in Appendix A.8.
>
> When we increase $\alpha$ to 1.0, the loss did not show significant fluctuations, but the cross-shot consistency slightly decreased.
>
> | Setting | Cross-shot DINO | Cross-shot CLIP |
> |:---|:---:|:---:|
> | $\alpha$=0.1 | 0.7284 | 0.8970 |
> | $\alpha$=1.0 | 0.7157 | 0.8851 |
>
>
> > **Q3: Dynamically adjusting the number of groups**
>
> Thank you for the thoughtful suggestion. Dynamically adjusting groups potentially enhances stability at large denoising timesteps. Since the principle of MoGA is to keep the design as simple and effective as possible, we put it in future work.
>
>
> > **Q4 and W4: Multimodal control evaluation**
>
> Following the reviewer’s suggestion, in the revised version we *add* CLIP scores to Tab. 1/2/3 and conduct an additional user study (see Appendix A.12).
>
>
> > **Q5: Resource requirements**
>
> For minute-long videos, MoGA with group size M=20 is trained on 256 Ascend 910B NPUs (TFLOPs roughly equal to A100; 64 GB memory). The total training time is about 50 hours.
>
> > **Q6: Broader applicability**
>
> Thank you for the suggestion. Since our paper focuses on video generation, we do not test other areas. In other areas such as LLMs, the main consideration is how to extend MoGA to causal attention for both training and inference.

---

> > ### Author Response · Authors · 2025-11-26
> > **Author Response to Reviewer ANEs (Part 2/2)**
> >
> > > **W1: Router design details**
> >
> > *How router collapse affects performance*: When the router collapses, the model degenerates into full attention. In this case, the training becomes 1.7× slower, with group size M=5 and a video duration of 30s.
> >
> > * *Dimension*: Since the router is a linear layer whose input and output are determined by the model dimension and the number of groups, it has no hyperparameter of dimension.
> >
> > * *Temperature*: Following the reviewer’s suggestion, we adjust the temperature. Different temperatures do not show clear difference.
> >
> >     | temperature | Cross-shot DINO | Cross-shot CLIP |
> >     |:---:|:---:|:---:|
> >     | T=1 | 0.7284 | 0.8970 |
> >     | T=0.5 | 0.7286 | 0.8946 |
> >
> > * *Regularization*: When we increase the group balancing loss weight to α = 1.0, the loss does not show significant fluctuations, but the cross-shot consistency slightly decreases.
> >
> >     | $\alpha$ | Cross-shot DINO | Cross-shot CLIP |
> >     |:---:|:---:|:---:|
> >     | 0.1 | 0.7284 | 0.8970 |
> >     |1.0 | 0.7157 | 0.8851 |
> >
> > > **W2: Compute/resource**
> >
> > We report the results on Ascend 910B NPUs (TFLOPs roughly comparable to NVIDIA A100; 64 GB memory): (1) single operator speed comparisons between MoGA (with STGA) and full attention (FlashAttention), and (2) end-to-end training/inference wall-clock time and memory (sequence parallel=8; duration=30 s).
> >
> > * **Operator Comparisons (Latency)**:
> >
> >     | Duration | Seq Len | Full Attention | MoGA (M=5) | Speedup (M=5) | MoGA (M=20) | Speedup (M=20) |
> >     |:---|:---:|:---:|:---:|:---:|:---:|:---:|
> >     | 5 s  | 31,200  | 40.06 ms  | 27.43 ms  | 1.46× | 21.58 ms  | 1.86× |
> >     | 10 s | 62,400  | 158.83 ms | 81.15 ms  | 1.96× | 62.24 ms  | 2.55× |
> >     | 20 s | 124,800 | 650.08 ms | 231.81 ms | 2.80× | 151.16 ms | 4.30× |
> >     | 30 s | 187,200 | 1,423.52 ms | 455.57 ms | 3.12× | 267.24 ms | 5.33× |
> >
> > * **End-to-End Comparisons**:
> >
> >     For 30s training dataset and Wan2.1-14B, the wall-clock times are measured below.
> >     | |  Full Attention | MoGA (M=5) | MoGA (M=20) |
> >     |---|:---:|:---:|:---:|
> >     | Training time (per iter) | 66.87 s | 38.76 s | 29.84 s |
> >     |  ↳ Speedup vs. Full | — | 1.72× | 2.24× |
> >     | Training RAM  | 48.1 GB | 49.8 GB | 49.6 GB |
> >     | Inference time (per step) | 40.21 s | 19.06 s | 16.55 s |
> >     |  ↳ Speedup vs. Full | — | 2.10× | 2.43× |
> >     | Inference RAM | 33.4 GB | 38.6 GB | 38.6 GB |
> >
> > > **W3: Data pipeline reproducibility**
> >
> > We thank the reviewer for the suggestion. In the revised version, we add a more detailed description of the data pipeline to Appendix A.13.

---

### Official Review · Reviewer_j5yM · 2025-10-31

**Soundness:** 3
**Presentation:** 3
**Contribution:** 3
**Rating:** 8
**Confidence:** 4

**Summary:**

This paper introduces Mixture-of-Groups Attention (MoGA), a novel and efficient sparse attention mechanism designed to overcome the computational bottleneck of generating long, high-resolution videos with diffusion transformers. Instead of relying on traditional block-wise estimation for sparsity, MoGA employs a lightweight, learnable "token router," inspired by Mixture-of-Experts, to assign each token to a specific group based on its semantics. Standard self-attention is then performed independently within each group, drastically reducing computational complexity while enabling effective long-range interactions. This global attention mechanism is complemented by a local Spatial-Temporal Group Attention (STGA) to ensure local continuity, and the model is trained on a custom-built data pipeline that produces minute-long, multi-shot video samples.

**Strengths:**

The paper's primary strength lies in its elegant and highly effective solution to the long-context problem. By replacing coarse block-level scoring with a precise, end-to-end token router, MoGA represents a conceptual advance over prior sparse attention methods. The approach is remarkably practical, as it is kernel-free and seamlessly integrates with existing high-performance technologies like FlashAttention and sequence parallelism. The experimental results are state-of-the-art and convincingly demonstrate the model's ability to generate coherent, minute-long, multi-shot videos, supported by comprehensive quantitative metrics and strong ablation studies that validate the design choices.

**Weaknesses:**

One potential point of discussion is the method's reliance on a powerful, pre-existing base model (Wan2.1) for fine-tuning, which makes it slightly difficult to isolate the gains of MoGA from the inherent capabilities of the foundation model. Additionally, the paper introduces an impressive and complex data pipeline for creating multi-shot training samples; the importance of this high-quality, specialized data to the final result is significant and could be considered a major contribution in its own right, perhaps underemphasized in the context of the attention mechanism.

**Questions:**

The token router is shown to learn meaningful semantic groupings in an unsupervised manner. Could the authors elaborate on the nature of these learned groups? For instance, do certain groups consistently specialize in specific visual concepts (e.g., one group for faces, another for backgrounds, another for dynamic motion), or are the groupings more abstract and context-dependent? Understanding this could provide deeper insights into the model's internal workings

---

> ### Author Response · Authors · 2025-11-26
> **Author Response to Reviewer j5yM**
>
> Thank you for the thoughtful feedback and valuable suggestions. Below we respond point-by-point.
>
> > **Q1: Semantics of Token Routing**
>
> We agree with the reviewer’s suggestion that the nature of the token router should be thoroughly analyzed. We add qualitative visualizations to inspect the properties of specific routing groups.
>
> Furthermore, we develop a *quantitative analysis tool* to measure whether the router assigns semantically related tokens to the same group. We discuss this below and integrate the methodology and results in Appendix A.9 and A.10.
>
> **Qualitative Visualizations**
>
> As shown in Fig. 3 and Fig. 15 of the revised version, the routers can exhibit attention to specific visual concepts. For example, the first group of routers in the 14th DiT block consistently focuses on face-related tokens across various samples.
>
> **Quantitative Analysis Tool**
>
> We treat the router’s grouping of tokens as a form of unsupervised segmentation. We then use SAM2[1] to obtain foreground masks for each frame as ground truth (GT), and treat the tokens aggregated by each group as predictions. For each GT mask, we match the group whose prediction mask achieves the highest Intersection over Union (IoU), and use the IoU as the metric.
> We use the average IoU of different prompts as the final result. The IoU measures the router’s ability to assign tokens of the same category to a single group.
>
> * **Train vs. Random**: We compare three routing methods: random assignment, a randomly initialized router, and the trained router. After training, specific groups achieve substantially higher IoU (28.6%) than both the random baseline (15.6%) and the randomly initialized router (18.5%).
>
>     | Methods |IoU (%)|
>     |:---|:---:|
>     | Random | 15.6|
>     |MoGA (Before Training)|18.5|
>     |MoGA (After Training)|28.6 |
>
> * **Different Timesteps**: We evaluate the IoU of the router at different sampling steps during inference. The router assigns tokens suboptimally (low IoU, ~17%) at the initial denoising step (t=999). As denoising progresses, the IoU quickly rises above 25% (t=931) and remains stable, gradually increasing to 28.6%, which indicates that MOGA is robust in the semantics of token assignments throughout the denoising process.
>
>     | Time Step | 999 | 956 | 931 | 884 | 853 | 499 | 438 | 364 | 272 | 155 |
>     |:---|:---:|:---:|:---:|:---:|:---:|:---:|:---:|:---:|:---:|:---:|
>     | IoU (%) | 17.6 | 22.2 | 25.9 | 25.1 | 24.8 | 27.8 | 27.6 | 27.3 | 28.0 | **28.6** |
>
> * **Different Blocks**: We evaluate the IoU of the router at different DiT blocks during inference. The middle DiT blocks exhibit relatively stronger semantic grouping ability (IoU up to 31.0%), which is consistent with prior work indicating that intermediate layers of DiT can capture high-level semantic features.
>
>     | Block Index | 1 | 5 | 10 | 15 | 20 | 25 | 30 | 35 | 38 |
>     |:---|:---:|:---:|:---:|:---:|:---:|:---:|:---:|:---:|:---:|
>     | IoU (%)| 19.2 | 18.4 | 31.0 | 23.6 | 26.6 | 28.9 | 28.0 | 30.6 | 24.5 |
>
>
> > **W1: One potential point of discussion is the method's reliance on a powerful, pre-existing base model (Wan2.1) for fine-tuning, which makes it slightly difficult to isolate the gains of MoGA from the inherent capabilities of the foundation model.**
>
> We appreciate the reviewer’s comment. We compare with baselines as fairly as possible to isolate the effect of the base models. All comparisons apply the same base model.
>
>
> > **W2: Additionally, the paper introduces an impressive and complex data pipeline for creating multi-shot training samples; the importance of this high-quality, specialized data to the final result is significant and could be considered a major contribution in its own right, perhaps underemphasized in the context of the attention mechanism.**
>
> Thank you for highlighting the value of our multi-shot data pipeline. We agree that high-quality data play a key role. In fact, our data pipeline primarily follows established community practices. Components such as OCR cropping [2, 3, 4], MLLM caption [2, 3, 4], AutoShot cutting [5, 6], and VQA filtering [2, 3, 4] are widely adopted across prior work.
>
> ref:
>
> [1] Ravi N, et al. Sam 2: Segment anything in images and videos.
>
> [2] Wan T, et al. Wan: Open and advanced large-scale video generative models.
>
> [3] Gao Y, et al. Seedance 1.0: Exploring the Boundaries of Video Generation Models.
>
> [4] Kong W, et al. Hunyuanvideo: A systematic framework for large video generative models.
>
> [5] Liu L, et al. Phantom: Subject-consistent video generation via cross-modal alignment.
>
> [6] Wang Y, et al. Internvideo2: Scaling foundation models for multimodal video understanding.

---

### Official Review · Reviewer_o1S4 · 2025-11-01

**Soundness:** 3
**Presentation:** 3
**Contribution:** 3
**Rating:** 6
**Confidence:** 5

**Summary:**

This paper introduce Mixture-of-Groups Attention (MoGA), an efficient sparse attention that uses a lightweight learnable token router to precisely match tokens without blockwise estimation. By semantics-aware routing, MoGA enables effective long-range interactions.

**Strengths:**

see Summary

**Weaknesses:**

The authors should provide between 20 to 50 video samples to better demonstrate the capabilities and limitations of their method. Additionally, for each prompt, it would be beneficial to include comparisons with other state-of-the-art methods. Ideally, there should be 3 to 5 comparison methods with corresponding videos for each prompt.

The authors should provide detailed results of the user study, including statistical analysis and user feedback. This will help in understanding how the proposed method performs in terms of user satisfaction and practical usability.

The paper only presents sub-scores from VBench, which provides a limited view of the overall performance.

The caption in Figure 3 should be more detailed. Otherwise, it is difficult to guess the meaning of the figure.

Why are there only experimental results for 1.3B in Table 2, but not for 14B? Similarly, Table 1 and Table 3 also lack these results.

The idea is very simple and clear, and it is easy to understand. However, the main difficulty and contribution may lie in the engineering implementation of the code. Since the author has not provided open-source code or code in the supplementary materials, I strongly suggest that the author provide the real Python code functions for Algorithm 1 for me to review.

Figure 6 only compares the FLOPs, but I believe that the actual inference time is much more important. This conceals two issues that I am very interested in:
1. In fact, the computational cost of Spatial-Temporal Group Attn is already quite large, and that of Mixture-of-Groups Attn is also significant. I think that under the scenario of short videos, the impact of these two designs on speed is minimal.
2. From an engineering implementation perspective, Spatial-Temporal Group Attn and Mixture-of-Groups Attn cannot be parallelized. Therefore, they will actually be slower due to serial computation.

**Questions:**

If the author's response is satisfactory to me, I would be willing to maintain a positive score.

---

> ### Author Response · Authors · 2025-11-26
> **Author Response to Reviewer o1S4**
>
> Thank you for the thoughtful feedback and valuable suggestions. Below we respond point-by-point.
>
> > **W1: More video samples**
>
> We additionally provide 20*3 video cases via the anonymous link (https://anonymous.4open.science/r/MoGA), including MoGA and the compared SoTA methods. The "Lora+Wan" subfolder (i.e. IC_LoRA+Wan) may not display its full content in the anonymous web. Please download the project to view it.
>
> > **W2: User Study**
>
> Following your suggestion, we conduct a user study. Each user scores generated videos along three dimensions (*i.e.*, video quality, consistency, and prompt following) with the Likert ratings. The study covers 10s and 30s multi-shot scripts and conducts a blind comparison among MoGA, EchoShot, and IC-LoRA+Wan. The evaluation results are as follows, and are included in Appendix A.12 of the revised version.
>
> | Metric (↑ better)            | EchoShot | IC-LoRA+Wan | MoGA |
> |:---|:---:|:---:|:---:|
> | Prompt Following             | 6.97     | 5.58        | 8.47 |
> | Video Quality    | 7.11     | 4.53        | 8.05 |
> | Consistency      | 6.37     | 5.03        | 8.26 |
>
>
> > **W3: More Evaluation Metrics**
>
> We thank the reviewer for the suggestion. The metrics used in our paper follow recent works on multi-shot long video generation (e.g., Mixture of Context and Long Context Tuning).
>
> Following the reviewer’s advice, we further include Overall Consistency and Temporal Flickering metrics from VBench, as well as Text2Video CLIP Score. We exclude some VBench sub-metrics that are less relevant to the current multi-shot long video setting, such as Color, Object Class. Please refer to Tab. 1/2/3 in the revised version.
>
>
> > **W4: The caption in Figure 3 should be more detailed.**
>
> Thank you for your suggestion. We provide a more detailed description in the caption of Fig. 3.
>
>
> > **W5: Why are there only experimental results for 1.3B in Table 2, but not for 14B? Similarly, Table 1 and Table 3 also lack these results.**
>
> We thank the reviewer for the suggestion. To ensure a fair comparison, we adopt backbones with the same scale as the baselines. In our experiments, we *default* to the best available open-source model at the time of experimentation, Wan2.1-14B (as shown in Tab. 1 and 3). However, since the method compared in Tab. 2, EchoShot, only provides a 1.3B checkpoint, we use a 1.3B backbone in Tab. 2 for fairness.
>
> Following the reviewer’s suggestion, in the revised version we *add* MoGA results for two model sizes (1.3B and 14B) to Tab. 1/2/3 for comprehensive comparison.
>
>
> > **W6: Code**
>
> In the revised version, we include the **key PyTorch pseudocode** details of MoGA in Appendix A.6 for reproducation.
>
> > **W7: Inference Time**
>
> We thank the reviewer for the suggestion. Below, we provide a comparison of the absolute operator runtimes for MoGA (with STGA) and full attention (FlashAttention), as well as end-to-end model inference times on Ascend 910B NPUs (TFLOPs roughly comparable to NVIDIA A100; 64 GB memory). These results are added to the revised version.
>
> * **Operator Comparisons (Latency)**:
>
>     | Duration | Seq Len | Full Attention | MoGA (M=5) | Speedup (M=5) | MoGA (M=20) | Speedup (M=20) |
>     |:---|:---:|:---:|:---:|:---:|:---:|:---:|
>     | 5 s  | 31,200  | 40.06 ms  | 27.43 ms  | 1.46× | 21.58 ms  | 1.86× |
>     | 10 s | 62,400  | 158.83 ms | 81.15 ms  | 1.96× | 62.24 ms  | 2.55× |
>     | 20 s | 124,800 | 650.08 ms | 231.81 ms | 2.80× | 151.16 ms | 4.30× |
>     | 30 s | 187,200 | 1,423.52 ms | 455.57 ms | 3.12× | 267.24 ms | 5.33× |
>
> * **End-to-End Comparisons (Inference)**:
>
>     The wall-clock inference times (per step) for different video duration (5s/10s/30s) on Wan2.1-14B are measured below.  Although MoGA is designed for long context, it still outperforms full attention on (*multi-shot*) short videos at the operator level. For longer video, the speed advantage of MoGA becomes more obvious.
>     | Setting | Full Attention | MoGA (M=5) | MoGA (M=20) |
>     |:---|:---:|:---:|:---:|
>     | 5s (SP=4)  | 3.63 s | 3.37 s |  3.08 s |
>     |  ↳ 5s Speedup vs. Full | — | 1.08× | 1.18× |
>     | 10s  (SP=8)  | 5.65 s | 4.06 s |  3.58 s |
>     |  ↳ 10s Speedup vs. Full | — | 1.39× | 1.58× |
>     | 30s (SP=8)  | 40.21 s | 19.06 s | 16.55 s |
>     |  ↳ 30s Speedup vs. Full | — | 2.10× | 2.43× |

---

> ### Comment · Reviewer_o1S4 · 2025-11-28
>
> I appreciate the authors’ thorough responses. They have addressed most of my concerns. I am more than happy to accept this work! I suggest the authors could refine the related work with some pre-training-based video generation work, such as:
> [1] Huang, et al. "M4V: Multi-Modal Mamba for Text-to-Video Generation." arXiv preprint arXiv:2506.10915 (2025).

---

> ### Author Response · Authors · 2025-11-28
> **Author Response to Reviewer o1S4**
>
> We sincerely thank the reviewer for the positive feedback and valuable suggestions, which significantly strengthen our paper. We also update the Related Work (Section 4.1) to include a discussion of M4V (highlighted in blue).

---

### Official Review · Reviewer_Hbw2 · 2025-11-10

**Soundness:** 3
**Presentation:** 3
**Contribution:** 3
**Rating:** 8
**Confidence:** 3

**Summary:**

This paper proposes Mixture-of-Groups Attention (MoGA), a new sparse attention mechanism designed to improve computational efficiency and scalability for end-to-end long video generation with diffusion transformers.  MoGA employs a lightweight learnable token router that dynamically assigns tokens into semantically coherent groups. Using MoGA, the authors present a video generation model that can produce minute-level, multi-shot, 480p videos at 24 FPS with a context length up to 580 K tokens. Experiments across multiple baselines (Wan2.1, MMDiT, EchoShot, IC-LoRA) demonstrate improved visual quality, subject and background consistency, and substantial reductions in FLOPs.

**Strengths:**

- The introduction of token-level routing as a “mixture-of-groups” attention mechanism is novel and intuitive. It replaces coarse block sparsity with a more fine-grained, data-driven grouping that potentially generalizes better across tasks.
- The paper convincingly shows MoGA’s ability to reduce attention complexity from O(N^2) to approximately O(N^2/M) while preserving quality. The claimed 1.7× training/inference speedup .
- The design is compatible with FlashAttention, sequence parallelism, and existing DiT frameworks.
- Across metrics such as subject/background consistency, aesthetic quality, and cross-shot CLIP/DINO similarity, MoGA consistently outperforms baselines, including full-attention models.

**Weaknesses:**

### Major
- My major concern is the novelty compared to the previous works. While MoGA's token router is inspired my MoE, and MoBA, it should be deeply analyzed the distinction between it and these works. The paper should clarify how fundamentally MoGA differs from the existing routing based methods, beyond the token level analysis.

- The benchmarks are built on top of the models like wan2.1/MMDiT and tested on internal datasets, where the authors integrate their MoGA module into the original architectures and train/fine-tune them on their internal dataset. Therefore, it is uncertain whether MoGA’s advantages generalize to other diffusion transformer backbones or unseen datasets.  While the paper provides an anonymous demo link, it lacks details about code or data release, which is critical for an academic paper submission claiming major efficiency improvements, also reproducibility.

### Others
- It could be better if the paper could provide formal analysis of stability, or the distribution of token assignments. "lightweight router" would be thoretically explained.

- It develops a multi-stage pipeline, which may be overly complicated (with steps like AutoShot, OCR, and LLM captioning),  should provide a experimental study on how much these steps actually improve performance, or a simpler baseline with a simplified pipeline. It would make the contribution clearer and more convincing.

**Questions:**

1. How is Mixture-of-Groups Attention fundamentally different from Mixture-of-Experts (MoE) or Mixture-of-Block Attention (MoBA/VMoBA) beyond operating tokens? Can you clarify what new insights or mechanisms MoGA introduces that are not already explored in MoBA (Lu et al., 2025) or Radial Attention (Li et al., 2025)?

2.What ensures that the lightweight router produces semantically meaningful groupings instead of arbitrary clusters? Have you analyzed how stable the routing assignments are across training iterations or input perturbations? Maybe could provide quantitative evidence (e.g., token entropy or similarity distributions) to support that MoGA learns meaningful semantic partitioning?

3. Have you tested MoGA on other architectures or datasets (e.g., Open-Sora, Pika, or VideoCrafter) to show generality beyond Wan/MMDiT? Do you expect MoGA to benefit smaller or lower-resolution models similarly, or is the gain limited to large-scale settings? It claims 1.7× speedup and reduced FLOPs; can you provide absolute runtime and memory usage numbers compared to full attention on the same hardware?

4. Please explain on reproducibility, e.g. release of the code, or ease of reimplementation. How can the efficiency be verified without access to the internal dataset and pipeline.

---

> ### Author Response · Authors · 2025-11-26
> **Author Response to Reviewer Hbw2 (Part 1/2)**
>
> Thank you for the thoughtful feedback and valuable suggestions. Below we respond point-by-point.
>
> > **Q1 & W1: Differences from related work**
>
> Below we clarify the distinctions from MoE, MoBA/VMoBA, and Radial-Attention. We incorporate these points into the Section 4.2 of the revised version.
>
> **MoE**:
> * Both MoE and MoGA leverage sparsity to reduce computation, but they *differ* in the objective and locus of sparsity. MoE scales **model parameters** by routing tokens to expert FFNs, *i.e.*, sparsity over parameters.
> * In contrast, MoGA scales with respect to **sequence length** by modifying attention and routing tokens into different attention groups, *i.e.*, sparsity over token interactions.
>
> **MoBA/VMoBA**:
> * MoBA selects important q–k *block* pairs via coarse-grained block-wise similarity maps. VMoBA extends this idea to video with tailored block structures and threshold-based selection. However, both methods are highly sensitive to the block size: smaller blocks improve performance but substantially increase the cost of coarse-grained similarity computation (as shown in Fig. 1 and theoretically supported by recent work [1]).
> * Our *insight* is to reinterpret the token routing as clustering and to *replace* explicit similarity-map computation with an end-to-end, learnable, and lightweight Router. Concretely, the router is a single linear layer whose columns serve as learnable cluster centroids, enabling efficient token-to-group assignment.
>
> **Radial-Attention**:
> * Radial-Attention uses a *fixed* sparse mask whose sparsity grows with the query–key distance. This injects a *hand-crafted inductive bias* that degrades over long sequences and impedes modeling of dynamic long-range consistency. In contrast, MoGA performs *semantic, distance-agnostic dynamic* token routing, which is crucial for long video generation, where semantically related frames can be arbitrarily far apart.
> * Moreover, the implementation of Radial-Attention requires an *O(N^2)* attention mask, which is prohibitive for long videos. For a one-minute video (~580K tokens), the mask alone would exceed 300 GB of GPU memory. MoGA avoids the mask-based design, making it practical at long context.
>
>
> > **Q2 & W3: Quantitative analysis of token routing**
>
> We agree with the reviewer's suggestion. Building on the original qualitative analysis (Fig. 3), we develop a **quantitative analysis tool** to measure whether the router of MoGA assigns semantically related tokens to the same group.
>
> Inspired by unsupervised segmentation, we use SAM2 [2] to obtain foreground masks for each frame as ground truth (GT), and treat the tokens aggregated by each group as predictions. For each GT mask, we match the group whose prediction mask achieves the highest Intersection over Union (IoU), and use the IoU as the metric. We use the average IoU of different prompts as the final result.
>
> We include the detailed methodology, quantitative results, and more visualization results in Appendix A.9 and A.10 of the revised version. Below we briefly analyze the main experimental findings.
>
> * **Train vs. Random**: We compare three routing methods: random assignment, a randomly initialized router, and the trained router. After training, specific groups achieve substantially higher IoU (28.6%) than both the random baseline (15.6%) and the randomly initialized router (18.5%), confirming learned semantic assignment.
>
>     | Methods |IoU (%)|
>     |:---|:---:|
>     | Random | 15.6|
>     |MoGA (Before Training)|18.5|
>     |MoGA (After Training)|28.6 |
>
> * **Different Time Step**: We evaluate the IoU of the router at different sampling steps during inference. The router assigns tokens suboptimally (low IoU, ~17%) at the initial denoising step (t=999). As denoising progresses, the IoU quickly rises above 25% (t=931) and remains stable, gradually increasing to 28.6%, which indicates that MoGA is robust in the semantics of token assignments throughout the denoising process.
>
>     | Time Step | 999 | 956 | 931 | 884 | 853 | 499 | 438 | 364 | 272 | 155 |
>     |:---|:---:|:---:|:---:|:---:|:---:|:---:|:---:|:---:|:---:|:---:|
>     | IoU (%) | 17.6 | 22.2 | 25.9 | 25.1 | 24.8 | 27.8 | 27.6 | 27.3 | 28.0 | **28.6** |
>
> * **Different Blocks**: We evaluate the IoU of the router at different  DiT blocks during inference. The middle DiT blocks exhibit relatively stronger semantic grouping ability (IoU up to 31.0%), which is consistent with prior work [3] indicating that intermediate layers of DiT can capture high-level semantic features.
>
>     | Block Index | 1 | 5 | 10 | 15 | 20 | 25 | 30 | 35 | 38 |
>     |:---|:---:|:---:|:---:|:---:|:---:|:---:|:---:|:---:|:---:|
>     | IoU (%)| 19.2 | 18.4 | 31.0 | 23.6 | 26.6 | 28.9 | 28.0 | 30.6 | 24.5 |
>
>
> ref:
>
> [1] Xiao G, et al. Optimizing Mixture of Block Attention.
>
> [2] Ravi N, et al. Sam 2: Segment anything in images and videos.
>
> [3] Yu S, et al. Representation alignment for generation: Training diffusion transformers is easier than you think.

---

> > ### Author Response · Authors · 2025-11-26
> > **Author Response to Reviewer Hbw2 (Part 2/2)**
> >
> > > **Q3 & W2: Test on other architectures or unseen datasets**
> >
> > We agree that validation on other architectures (*e.g.*, autoregressive models) is a reasonable direction for future work. Due to the tight rebuttal timeline, we have not conducted these validations yet.
> >
> > In principle, MoGA is a general sparse attention suitable for long-sequence modeling. Current empirical evidence demonstrates MoGA’s generalization across a range of backbones, including Wan and MMDiT architectures (two prevailing video generation architectures).
> >
> > In addition, the test dataset in Tab. 2 is an *unseen* dataset sourced from EchoShot, whose distribution differs from our training data. Despite this shift, our model produces strong generations, demonstrating MoGA’s robust generalization ability.
> >
> >
> > > **Q3: Smaller or lower-resolution models**
> >
> > We expect MoGA also benefits smaller or lower-resolution models. Our experiments show consistent improvements across multiple scales (including a 1.3B small model). Under the same VAE, lower-resolution inputs correspond to shorter sequence lengths. Notably, we evaluate MoGA across various sequence lengths (*i.e.*, 30k–580k) in our experiments.
> >
> >
> > > **Q4: Runtime and memory usage**
> >
> > We report results on the Ascend 910B NPU (TFLOPs roughly comparable to NVIDIA A100; 64 GB memory): (1) single operator speed comparisons between MoGA (with STGA) and full attention (FlashAttention), and (2) end-to-end training/inference wall-clock time and memory (Sequence Parallel=8; duration=30 s).
> >
> > * **Operator Comparisons (Latency)**:
> >
> >     | Duration | Seq Len | Full Attention | MoGA (M=5) | Speedup (M=5) | MoGA (M=20) | Speedup (M=20) |
> >     |:---|:---:|:---:|:---:|:---:|:---:|:---:|
> >     | 5 s  | 31,200  | 40.06 ms  | 27.43 ms  | 1.46× | 21.58 ms  | 1.86× |
> >     | 10 s | 62,400  | 158.83 ms | 81.15 ms  | 1.96× | 62.24 ms  | 2.55× |
> >     | 20 s | 124,800 | 650.08 ms | 231.81 ms | 2.80× | 151.16 ms | 4.30× |
> >     | 30 s | 187,200 | 1,423.52 ms | 455.57 ms | 3.12× | 267.24 ms | 5.33× |
> >
> > * **End-to-End Comparisons**:
> >
> >     For 30 s training dataset and Wan2.1-14B, the wall-clock times are measured below.
> >     | |  Full Attention | MoGA (M=5) | MoGA (M=20) |
> >     |---|:---:|:---:|:---:|
> >     | Training time (per iter) | 66.87 s | 38.76 s | 29.84 s |
> >     |  ↳ Speedup vs. Full | — | 1.72× | 2.24× |
> >     | Training RAM  | 48.1 GB | 49.8 GB | 49.6 GB |
> >     | Inference time (per step) | 40.21 s | 19.06 s | 16.55 s |
> >     |  ↳ Speedup vs. Full | — | 2.10× | 2.43× |
> >     | Inference RAM | 33.4 GB | 38.6 GB | 38.6 GB |
> >
> >
> > > **Q5: Reproducibility**
> >
> > In the revised version, we include a detailed **key PyTorch pseudocode** of MoGA in Appendix A.6 for reproduction.
> >
> >
> > > **W4: Data pipeline**
> >
> > Thank you for the suggestion. Our data pipeline primarily follows established community practices. Components such as OCR cropping [4, 5, 6], MLLM caption [4, 5, 6], AutoShot cutting [7, 8], and VQA filtering [4, 5, 6] are widely adopted across prior work.
> >
> > ref:
> >
> > [4] Wan T, et al. Wan: Open and advanced large-scale video generative models.
> >
> > [5] Gao Y, et al. Seedance 1.0: Exploring the Boundaries of Video Generation Models.
> >
> > [6] Kong W, et al. Hunyuanvideo: A systematic framework for large video generative models.
> >
> > [7] Liu L, et al. Phantom: Subject-consistent video generation via cross-modal alignment.
> >
> > [8] Wang Y, et al. Internvideo2: Scaling foundation models for multimodal video understanding.

---

### Author Response · Authors · 2025-11-27
**Author General Response**

We sincerely thank all four reviewers for their thoughtful and constructive feedback. The uniformly positive feedback greatly motivates us. We are pleased that the reviewers consistently recognize the novelty and intuitiveness of our proposed MoGA.

We address the comments in the individual responses and update the paper accordingly, with key changes highlighted in blue. In summary, the main revisions are:

1. **[Reviewer Hbw2: Distinctions from related work]** Add clearer distinctions from related methods (*e.g.*, MoE, MoBA, VMoBA, Radial-Attention) in Sections 2.2 and 4.2.

2. **[Reviewer o1S4: More clear description]** Provide an improved caption explanation of Fig. 3.

3. **[Reviewer o1S4 and ANEs: Additional metrics]** Expand evaluation in Tab. 1/2/3 with additional metrics (*e.g.*, Text2Video CLIP Score) and the results of MoGA 1.3B/14B.

4. **[Reviewer Hbw2, o1S4 and ANEs: Detailed code]** Provide a detailed PyTorch implementation of MoGA in Appendix A.6.

5. **[Reviewer ANEs: Hyperparameters ablation]** Include additional ablation analysis on hyperparameters of the router (*e.g.*, temperature, loss weight) in Appendix A.7 and A.8.

6. **[Reviewer Hbw2, j5yM and ANEs: Token router analyses]** Add (more) qualitative visualizations and quantitative analyses of token routing in Appendix A.9 and A.10.

7. **[Reviewer Hbw2, o1S4 and ANEs: Runtime and memory usage]**  Report runtime and memory usage in Appendix A.11.

8. **[Reviewer o1S4 and ANEs: User study]** Present user study in Appendix A.12.

9. **[Reviewer ANEs: Detail of data pipeline]**  Provide more details of the data pipeline in Appendix A.13.

---

### Meta-Review · Area_Chair_C4Mv · 2026-01-03

**Summary:**

The paper proposes MoGA, an efficient sparse attention mechanism for long video generation. It uses a router to assign tokens to specific groups, which enables effective modeling of long contexts.

The reviewers agree that the idea of MoGA is novel and intuitive. MoGA outperforms both sparse and full-attention baselines, and the ablations are comprehensive. The proposed MoGA has shown improvement for open-source video models, and it could generalize to other long-sequence tasks.

There are also several weaknesses pointed out by the reviewers, which have been addressed during the rebuttal phase. Therefore, I would recommend acceptance of this work. I encourage the authors to incorporate reviewers' suggestions in their next version.

**Reviewer Concerns:**

Most of the major concerns are addressed by the rebuttal.

**Reviewer Scores:**

8, 6, 8, 6 --> 8, 8, 8, 6

---

### Decision · Program_Chairs · 2026-01-26

Accept (Poster)